# CPK1 activates CNGCs through phosphorylation for Ca²⁺ signaling to promote root hair growth in Arabidopsis

Meijun Zhu [1,2], Bo-Ya Du [1,2], Yan-Qiu Tan [1], Yang Yang [1,2], Yang Zhang [1,2] & Yong-Fei Wang [1,2] ✉

Cyclic nucleotide-gated channel 5 (CNGC5), CNGC6, and CNGC9 (CNGC5/6/9 for simplicity) control Arabidopsis root hair (RH) growth by mediating the influx of external Ca²⁺ to establish and maintain a sharp cytosolic Ca²⁺ gradient at RH tips. However, the underlying mechanisms for the regulation of CNGCs remain unknown. We report here that calcium dependent protein kinase 1 (CPK1) directly activates CNGC5/6/9 to promote Arabidopsis RH growth. The loss-of-function mutants *cpk1-1, cpk1-2, cngc5-1 cngc6-2 cngc9-1 (shrh1/short root hair 1)*, and *cpk1 shrh1* show similar RH phenotypes, including shorter RHs, more RH branching, and dramatically attenuated cytosolic Ca²⁺ gradients at RH tips. The main CPK1-target sites are identified as Ser20, Ser27, and Ser26 for CNGC5/6/9, respectively, and the corresponding alanine substitution mutants fail to rescue RH growth in *shrh1* and *cpk1-1*, while phospho-mimic versions restore the cytosolic Ca²⁺ gradient at RH apex and rescue the RH phenotypes in the same Arabidopsis mutants. Thus we discover the CPK1-CNGC modules essential for the Ca²⁺ signaling regulation and RH growth in Arabidopsis.

The plant root system is essential for anchorage and the acquisition of nutrient ions and water, and forms an important interface with soil microbes[1–3]. Root hairs (RHs), the tubular-shaped structures derived from the outgrowth of root epidermal cells, play essential roles in these processes[1–3]. RHs adhere to the surface of soil particles and release adhesive exudates that aid root penetration into the soil[4–8]. It has been also reported that the roots of English ivy (*Hedera helix*) exude a yellowish mucilage that promotes the capacity of the plant to climb vertical surfaces, and RHs play an important role in the plant climbing[9]. RHs greatly increase the area of the root-soil interface and so play a major role in the uptake of nutrients and water, and, along with root system architecture, largely determine the volume of the root-soil interface that plants can exploit. An early study reported that the RH interface of a two-week old winter rye (*Secale cereale*) plant is approximately 4321 square feet (401 square meter), which is almost twice that of the remaining root surface[3,10]. The diameter of the RH cylinder around the root is approximately ten times larger than that of

the root, and thus the volume of the RH cylinder is about 100 times larger than that of the root[3,10].

The essential role that RHs play in the uptake of water and nutrient ions from the soil, including the macronutrients nitrate, potassium, phosphate and calcium, and micronutrients, is indisputable[4,6,11,12]. This is reflected by the strong effect of ion nutrient availability (deficiency or sufficiency) on RH density, RH length, and plant biomass[13–18]. Moreover, impairment of RH growth results in the decreased uptake of nutrient ions and water, and consequently reduces the tolerance of plants to abiotic stresses and results in the reduction of plant biomass[4,6,11,12,19]. Thus, the ability of the root system to take up water and important nutrient ions, especially immobile ions such as phosphate, can be significantly increased by promoting RH growth. This is extremely important for the breeding of new crop cultivars with enhanced performance in less fertile soils[3].

RHs are distributed in the differentiation zone of the root. In Arabidopsis, the diameter of RHs is approximate 10 μm, and RHs can

[1]State Key Laboratory of Plant Trait Design, CAS Center for Excellence in Molecular Plant Sciences, Chinese Academy of Sciences (CAS), Shanghai 200032, China. [2]University of Chinese Academy of Sciences, Shanghai 200032, China. ✉e-mail: yfw@cemps.ac.cn

grow to approximate 1 mm in length at rate of more than 1 μm per min at a fixed angle of about 85° relative to the root surface[20]. The development of RHs can be roughly categorized into three steps: cell-fate determination of the root epidermal cells, the initiation of RHs, and subsequent elongation through tip growth. The epidermal cells of roots are determined to be either hair (H) cells (trichoblasts) or non-hair (N) cells (atrichoblasts), and this process follows a strict cell position-dependent pattern. H cells are present over the intercellular space between two underlying cortex cells, and N cells are present over a single cortex cell[20]. The cell-position-dependent pattern for cell-fate determination of root epidermis is mainly controlled genetically by an intrinsic regulating network[21,22], and GLABRA2 (GL2), a homeodomain transcription factor protein, functions as a central component of the regulating network[21,22]. GL2 is mainly expressed in the epidermal cells adopting an N cell-fate[21], and loss of GL2 results in the production of excessive RHs[21,22]. Thus GL2 is a positive regulator of N cell-fate differentiation and a negative regulator of H cell-fate differentiation in the epidermis. The expression of GL2 is positively regulated by a number of upstream transcription factors, including TRANSPARENT TESTA GLABRA (TTG)[23,24], the basic helix-loop-helix (bHLH) transcription factors GLABRA3 (GL3) and ENHANCER OF GLABRA3 (EGL3)[25,26], and the R2R3-MYB transcription factors WEREWOLF (WER) and MYB23[27,28]. WER, GL3, EGL3, and TTG form a complex with WER as the central component. By contrast, CAPRICE (CPC) and its redundant partners positively regulate H cell-fate differentiation by negatively regulating the WER-GL3-EGL3-TTG pathway. CPC binds to WER in a competitive manner to prevent the binding of GL3/EGL3 to WER[29–32]. Moreover, CPC is capable of traveling from one epidermal cell to another to repress the N cell-fate[32,33]. In addition to this network governing epidermal cell-fate determination, a number of other genes are also involved in this process, acting either independently or through interactions with the network components[20].

Once H cell fate is determined, RH growth is initiated. The initiation of RHs is a process of cell-wall loosening and bulge/swelling formation[34–36]. The acidification of the apoplastic space and the production of reactive oxygen species (ROS) facilitate cell wall loosening for the RH initiation[34–36]. ROS production during RH initiation is mainly catalyzed by the RESPIRATORY BURST OXIDASE HOMOLOG C/ROOT HAIR DEFECTIVE2 (RBOHC/RHD2)[37]. The ROS then triggers external $Ca^{2+}$ influx by activating the $Ca^{2+}$-permeable channel Annexin 1 to elevate the cytosolic $Ca^{2+}$ concentration in the bulges[38]. The accumulation of ROS also facilitates the softening of the cell wall in the bulges to allow the turgor pressure-driven expansion of the bulge for RH tip growth[39,40]. Thus ROS and $Ca^{2+}$ form a positive regulatory loop for the RH initiation and tip growth[41].

RHs grow uni-directionally by depositing cell wall and plasma membrane (PM) materials at their apex via exocytosis, and, similar to pollen tubes, RHs use turgor pressure as the driving force to expand their volume and surface at RH tips[42,43]. This process is referred to as tip growth. The tip growth of RHs is regulated by phytohormones, including auxin, ethylene, jasmonate (JA), abscisic acid (ABA), gibberellin (GA), strigolactones (SL), cytokinin (CK), and brassinosteroids (BR)[44]. RH growth is also affected by changes in environmental conditions, such as the availability of nutrients and water, and interactions with soil microbes[3]. These factors regulate RH tip growth through intrinsic regulatory machinery through changes in $Ca^{2+}$, ROS, pH, the cytoskeleton, ROPs etc.[44]

External $Ca^{2+}$ influx occurs during RH initiation[37], and the influx of $Ca^{2+}$ at the bulges of trichoblasts forms a $Ca^{2+}$ gradient at their apex, where the cytosolic $Ca^{2+}$ concentration can reach about 1 μM and decreases to about 0.1 μM at the base of RHs[45,46]. The $Ca^{2+}$ gradient at the RH apex oscillates with a frequency of 2-4 peaks per min, as does the tip growth of RHs, but the $Ca^{2+}$ oscillation lags the RH growth oscillation by a few seconds[47,48]. The cytosolic $Ca^{2+}$ gradient at the RH apex functions as a key regulator of both RH tip growth and

orientation and eventually vanishes when tip growth ceases[45,48,49]. During RH growth, the cytosolic $Ca^{2+}$ oscillation is highly coordinated with and interacts to other regulating factors, including ROS, pH, ROPs, and cytoskeleton, suggesting a central role for $Ca^{2+}$ as a signal in RH growth regulation. Several ROPs function as PM-localized switches for the tip growth of RHs and pollen tubes[50–53]. The activated ROPs are involved in RH growth through their regulation of exocytosis and endocytosis, changes in the cytoskeleton, ROS production, and $Ca^{2+}$ signaling[39,54–58]. ROS are known as signaling molecules, and polarized ROS production is required for RH tip growth[37]. The knockout mutant rhd2/rbohc failed to accumulate ROS, and developed only very short RHs[37]. Further analysis revealed that the ROP2-RHD2-ROS cascade is involved in RH tip growth regulation by activating PM $Ca^{2+}$ channels and external $Ca^{2+}$ influx[37,39]. The elevation of cytosolic $Ca^{2+}$ can also stimulate ROS production through the activation of RBOHC/RHD2[41]. Thus, ROS and $Ca^{2+}$ form a positive regulatory loop in RHs. F-actin and microtubules are involved in the movement of organelles and the nucleus along the cytoskeleton for their proper positioning to support RH growth[59–62]. $Ca^{2+}$ signaling regulates cytoskeletal dynamics via $Ca^{2+}$-binding proteins, and the elevation of cytosolic $Ca^{2+}$ facilitates the depolymerization of F-actin filaments and microtubules[63].

Despite the central role of the cytosolic $Ca^{2+}$ in RH growth, the underlying mechanism by which the $Ca^{2+}$ gradient is retained and regulated for RH growth is still largely unknown. It has been well established that external $Ca^{2+}$ influx is mainly focused at RH tips rather than the shank area, and consequently the cytosolic $Ca^{2+}$ gradient is established and retained at RH apex for RH growth[45,64–66]. Thus, the $Ca^{2+}$ channels responsible for the external $Ca^{2+}$ influx into RHs were thought to be localized mainly at RH tips. Later on, electrophysiological analysis provided evidence for the presence of the PM $Ca^{2+}$-permeable channels at RH tips[37,67]. $Ca^{2+}$-permeable channels belong to a few families, including cyclic nucleotide-gated channels (CNGCs), glutamate receptor-like channels (GLRs), and Annexins, all of which are candidates to mediate RH tip growth. However, the PM $Ca^{2+}$ channels involved in RH growth remained undiscovered for decades. Recently, three CNGCs, namely CNGC5, CNGC6, and CNGC9 (CNGC5/6/9 for simplicity), were revealed to be the PM $Ca^{2+}$ channels essential for the tip growth of Arabidopsis RHs, and the triple mutant cngc5-1 cngc6-2 cngc9-1 designated as short root hair1 (shrh1) and the quadruple mutant cngc5-1 cngc6-2 cngc9-1 cngc14-1 designated as short root hair 2 (shrh2) showed similar defects in RH growth[50,51]. The subcellular localization of the CNGC5/6/9 at RH tips has been clearly demonstrated[50,51]. These studies demonstrate that the CNGCs are the main $Ca^{2+}$ channels that mediate RH growth[48,49]. Another study revealed that CNGC14 is essential for the touch sensing of RHs because the knockout mutant cngc14 showed RH defects only in a condition that RHs grow in solid medium[68]. However, it is still unknown how these CNGCs are regulated to control the tip-focused $Ca^{2+}$ gradient and RH tip growth.

The Arabidopsis genome contains 20 CNGC members. Each CNGC has six transmembrane domains (S1-S6) with a pore-forming loop between S5 and S6, a short N terminus, and a long C terminus with a cyclic nucleotide-binding domain (CNBD) and a calmodulin (CaM)-binding domain (CaMBD), similar to their mammalian orthologs cyclic nucleotide-gated (CNG) channels[69,70]. Both mammalian CNGs and plant CNGCs can form PM $Ca^{2+}$ channels as either homo- or hetero-tetramers[70–72]. Mammalian CNGs are activated by the binding of cyclic nucleotides (cAMP and cGMP)[70], but no activation of plant CNGCs by cyclic nucleotides was observed in Xenopus laevis oocytes[72]. Thus, it appears that the cyclic nucleotides may not be upstream regulators of CNGCs. On the other hand, protein phosphorylation/dephosphorylation is a general regulatory mechanism for regulating the activity of diverse ion channels. For instance, the $Ca^{2+}$-permeable channels Annexin1 and CNGCs can be activated by OPEN STOMATA 1(OST1)-mediated phosphorylation in response to cold and drought

stress/abscisic acid (ABA), respectively[73,74]. The anion channels slow anion channel associated 1 (SLAC1), SLAC1 homologues (SLAHs), and K$^+$ channels from Shaker family can be phosphorylated by OST1 and calcium-dependent protein kinase (CPK/CDPK) in guard cells for stomatal movement regulation[75–78]. Thus, the CNGCs are very likely regulated by kinase- and phosphatase-mediated protein phosphorylation and dephosphorylation, and the kinases from diverse protein kinase families, including CPK/CDPK, calcineurin B-like (CBL) protein-interacting protein kinase (CIPK), SNF1-related protein kinase 2 (SnRK2), and mitogen-activated protein kinase (MAPK) cascade, are attractive candidates.

In this research, we screened an Arabidopsis protein kinase mutant library for RH mutants. This led to the identification of the loss-of-function mutants *cpk1-1* and *cpk1-2* as having *shrh1*-like RH defects. We found that CPK1 positively regulates RH growth by activating CNGC5/6/9 through protein phosphorylation at a conserved serine within the N terminal region of the three CNGCs. We found that the expression of phospho-mimic CNGC5/6/9 with an S-to-D point mutation at the main CPK1-target sites is sufficient to rescue the RH phenotypes of *cpk1* and *shrh1*, revealing a CPK1-CNGC signaling module that is essential for RH growth, and shedding light on the molecular mechanism underlying the regulation of cytosolic Ca$^{2+}$ in this process.

## Results

### CPK1 is required for RH growth in Arabidopsis

Cyclic nucleotides are unlikely to be upstream regulators of CNGCs because they are unable to activate CNGCs in *Xenopus leavis* oocytes[72]. We then focused on protein kinases as potential regulators of CNGCs in RHs.

To determine whether any kinases are essential for RH growth, we screened a kinase mutant library that contains mutants from most protein kinase families, including CPKs, SnRK2s, MAP kinases, and CBL-CIPKs, by analyzing RH phenotypes (Supplementary Table 1). Columbia (Col-0) wild-type plants and the triple mutant *shrh1*, the latter which shows significant defects in RH growth[48], were used as controls. Four-day old seedlings grown in plates containing solid half-strength Murashige and Skoog (1/2 MS) medium were analyzed. We performed optical sectioning under bright field, and each set of sectioning images were merged into a single 2-D image. Most RHs growing at different angles and layers can be observed clearly in the merged 2-D images, so we used them for RH phenotype analysis. We found that the T-DNA insertion mutants *cpk1-1* and *cpk1-2* showed significant defects in RH growth, including shorter RHs (Fig. 1a–c) and more RH branching (Fig. 1b, d), but their RH density was normal (Fig. 1a, e), compared to the wild type. Interestingly the RH-related phenotypes in *cpk1-1* and *cpk1-2* were similar to that of *shrh1* (Fig. 1a–e).

To monitor RH growth, we performed a time-lapse analysis for no less than 300 min. The data showed that the RHs of *cpk1-1* grew at a significantly lower speed (Supplementary Fig. 1a, b; Fig. 1f), and also ceased growth earlier (Supplementary Fig. 1a, b; Fig. 1g), compared to that of wild type. These phenotypes resemble the phenotypes of *shrh1* as previously reported[48].

We conducted RT-qPCR experiments, and the results showed that *CPK1* was knocked out in the mutants *cpk1-1* and *cpk1-2* (Supplementary Fig. 2a). We generated complemented (COM) lines by expressing wild type *CPK1* under its native promoter in the mutant *cpk1-2* background, and two *COM* lines *pCPK1-COM1* (*CPK1 COMPLEMENTATION 1 under CPK1 native promoter in cpk1-2)* and *pCPK1-COM5* were selected for further analysis. The RT-qPCR data showed that the *CPK1* was expressed in *pCPK1-COM1* and *pCPK1-COM5* at a level similar to that of wild type plants (Supplementary Fig. 2a). RH phenotype analysis showed that the defects in RH growth were successfully rescued in *pCPK1-COM1* and *pCPK1-COM5* compared to *cpk1-2* and wild type (Fig. 1a, c, d). We further generated overexpression (*OE*) lines by overexpressing wild type *CPK1* under a *Ubiquitin 10* (*UBQ10*) promoter

in the background of the wild type plants, and two *OE* lines *CPK1-OE6* (*CPK1 OVEREXPRESSION 6 in wild type background*) and *CPK1-OE13* were selected for RH phenotype analysis. Results from RT-qPCR showed that the expression of *CPK1* in the two *OE* lines was increased by approximately 7-10 fold compared to that of wild type plants (Supplementary Fig. 2a). We analyzed the RH phenotypes, and observed significant longer RHs in the two *OE* lines compared to that in the wild type (Fig. 1a, c). We rarely observed branching RHs (Fig. 1d), and the RH density was not obviously altered (Fig. 1a, e), compared to that of wild type plants.

To further analyze the functions of CPK1 in RHs, we generated transgenic Arabidopsis lines by expressing wild type *CPK1* under a RH-specific promoter *EXPANSIN A7* (*EXPA7*)[79] in the background of *cpk1-1*. Two transgenic lines *pEXPA7-COM13* (*CPK1 COMPLEMENTATION 13 under the promoter pEXPA7 in the background of cpk1-1*) and *pEXPA7-COM14* were selected for further experiments. RT-qPCR analysis was performed. The data showed that the *CPK1* was expressed in *pEXPA7-COM13* and *pEXPA7-COM14* in a level similar to that in the wild type plants (Supplementary Fig. 2a). We analyzed the RH phenotypes in the transgenic lines, and found that the RH length was restored to levels similar to that of wild type plants (Fig. 1a, c), the RH branching rates were dramatically reduced from more than 20% in *cpk1-1* to less than 0.5% in the two transgenic lines (Fig. 1d), and the RH density was not obviously altered in the *pEXPA7-COM13* and *pEXPA7-COM14* lines compared to *cpk1-1* and the wild type (Fig. 1a, e). These results together suggest that CPK1 plays an important role in RH growth.

### CPK1 is highly expressed in RHs, and CPK1 is localized in the periphery of RHs

To investigate the expression profile of *CPK1* in Arabidopsis, we generated transgenic Arabidopsis lines containing the *GUS* gene under the control of the native promoter of *CPK1* in the wild type background. GUS staining results showed that *CPK1* is highly expressed in RHs and the xylem of primary roots (Supplementary Fig. 3a).

A transgenic line, *pEXPA7-COM13* with eGFP fused to the C terminus of CPK1, was generated to study the subcellular localization of CPK1 in RHs. Strong eGFP fluorescence in the apexes of most RHs was observed (Supplementary Fig. 3b). We further monitored the subcellular localization of CPK1 in RHs, and the optical sections of individual RHs with 2 μm sectioning steps showed that CPK1 was mainly distributed in the periphery of RH cells, especially at the RH apex (Supplementary Fig. 3c), consistent with a potential role in RH growth.

### CPK1 functions upstream of CNGC5/6/9 for RH growth

We hypothesized that CPK1 may function as a direct upstream regulator of the CNGCs in RHs. To test this hypothesis, we generated *cpk1-1 cngc5-1*, *cpk1-1 cngc6-2*, and *cpk1-1 cngc9-1* double mutants and analyzed their RH phenotype. We observed strong defects of RH growth in the *cpk1-1* mutant (Supplementary Fig. 4a–c), while the single mutants *cngc5-1*, *cngc6-2*, and *cngc9-1* had wild type-like RHs (Supplementary Fig. 4a–d). The wild type-like RH growth observed in the three single *cngc* mutants is consistent with a previous report[48]. However, we observed *cpk1-1*-like defects of RH growth in the *cpk1-1 cngc5-1*, *cpk1-1 cngc6-2*, and *cpk1-1 cngc9-1* double mutants, including shorter RHs and higher RH branching frequencies (Supplementary Fig. 4a–c). The RH density was not obviously altered in the double mutants compared to the wild type (Supplementary Fig. 4a, d).

We next generated a *cpk1 shrh1* quadruple mutant by crossing the single mutant *cpk1-1* to the triple mutant *shrh1* (*cngc5-1 cngc6-2 cngc9-1*)[48]. We analyzed the RH phenotypes, and found that *cpk1-1*, *shrh1*, and *cpk1 shrh1* showed similarly strong RHs phenotypes with shorter RHs and more branches than wild type (Fig. 2a–c). Like the *cpk1-1* and *shrh1* mutants, the RH density was not obviously altered in the quadruple mutant *cpk1 shrh1* (Fig. 2a, d). This data demonstrates that the effects of the mutations in CPK1 and CNGC5/6/9 on RH

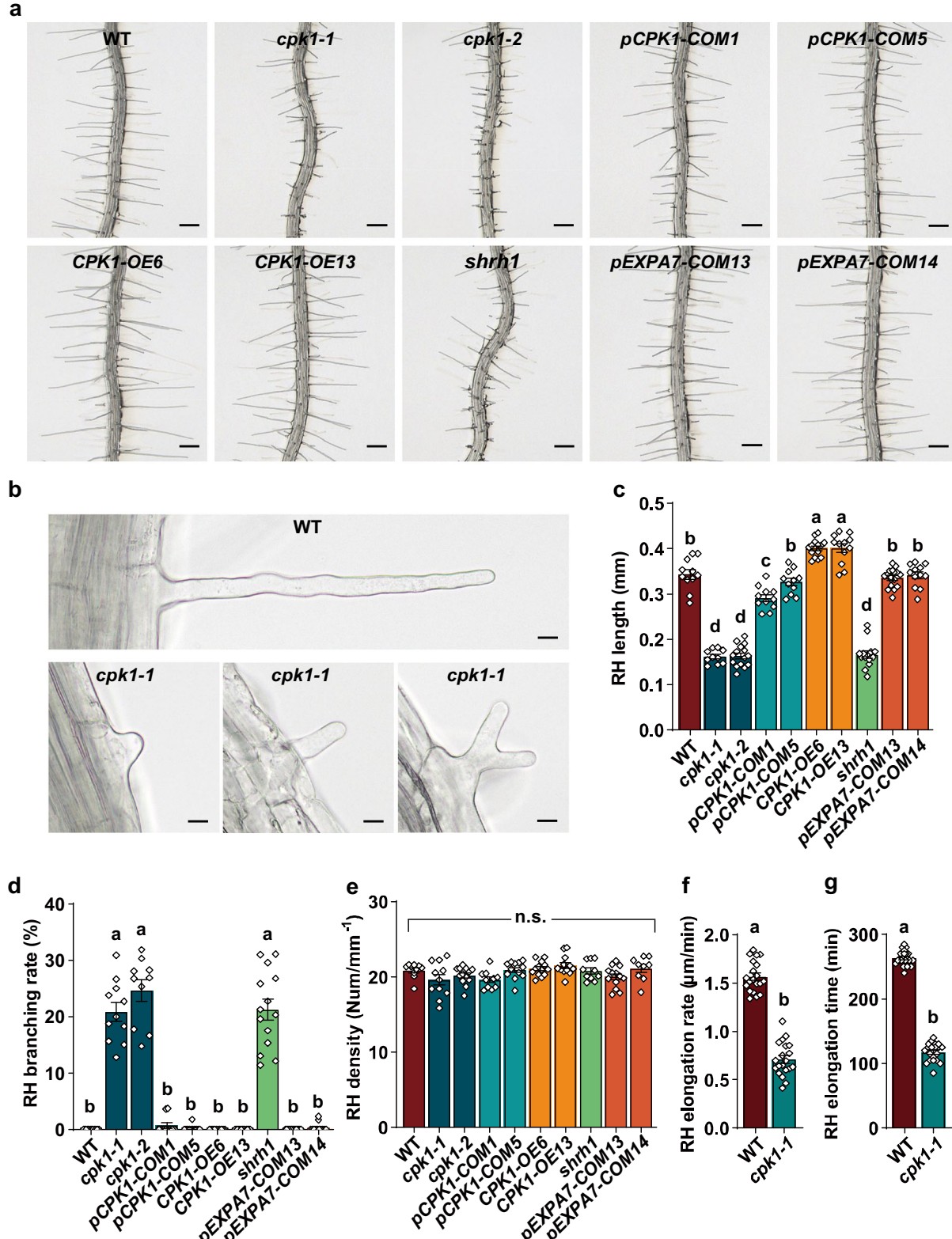

growth are not additive, suggesting that CPK1 and CNGC5/6/9 function in the same signaling pathway in Arabidopsis RHs.

It was previously shown that the Arabidopsis *CNGC9-OE42* line overexpressing *CNGC9* under the *UBQ10* promoter in the wild type Arabidopsis background conferred a longer RH phenotype[48]. As described above, we found that overexpression of *CPK1* under the *UBQ10* promoter conferred a similar phenotype (Fig. 1a, c)[48]. To study whether CPK1 functions upstream of CNGC5/6/9 in RH growth, we generated transgenic lines expressing *CPK1* under the *UBQ10* promoter in the background of *shrh1*, and one transgenic line designated as *CPK1-OE3* (*shrh1*) (*CPK1 OEVEREXPRESSION 3 in the background of shrh1*) was selected for further study. We also generated transgenic lines by overexpressing *CNGC9* under the *UBQ10* promoter in the background of *cpk1-1*, and one transgenic line was selected and designated as *CNGC9-OE1* (*cpk1-1*) (*CNGC9 OVEREXPRESSION 1 in the background of cpk1-1*). RT-qPCR data verified the over-expression of

**Fig. 1 | CPK1 is required for RH growth in Arabidopsis.** 4-5 day-old seedlings growing in Petri dishes were used for RH phenotype analysis. A set of optical section images of the RHZ of the seedlings with a 22.5 μm sectioning step and a sectioning speed of 5 steps/sec were captured under a stereo microscope at room temperature (25 ± 1 °C), and the set of images were automatically and immediately merged into a 2-D picture for each Petri dish. The 2-D pictures were used for RH phenotype analysis. For the time-lapse analysis of RH elongation rate and growth time, the optical sectioning was performed every min for 300 min for each dish, and the merged 2-D photos were used to analyze the RH elongation rate and time. **a, b** Typical 2-D photos derived from optical sectioning images of the RHZ of seedlings (**a**) and individual RHs showing the typical RH shapes (**b**). **c–g** Statistical

analyses of RH length (**c**), RH branching rates (**d**), RH density (**e**), RH elongation rate (**f**), and RH elongation time (**g**). Scale bars, 0.2 mm in (**a**), and 10 μm in (**b**). **c–e** The numbers of biologically independent roots with approximate 50 RHs per root examined in this study are 11, 14, 10, 15, 11, 11, 14, 12, 17, and 12 for RH length (**c**), 12, 14, 11, 10, 10, 11, 10, 13, 15, and 12 for RH branching (**d**), and 10, 10, 11, 13, 10, 12, 13, 12, 13, and 9 for RH density (**e**), for the 10 Arabidopsis lines as shown from left to right in each panel. **f, g** $n = 20$ roots for both WT and *cpk1-1*. Samples with different letters are significantly different with $P < 0.05$ (one-way ANOVA) in (**c–e**), and $P < 0.01$ (two-tailed Student's $t$-test) in (**f, g**). Data are presented as means ± SEM. Source data are provided as a Source Data file.

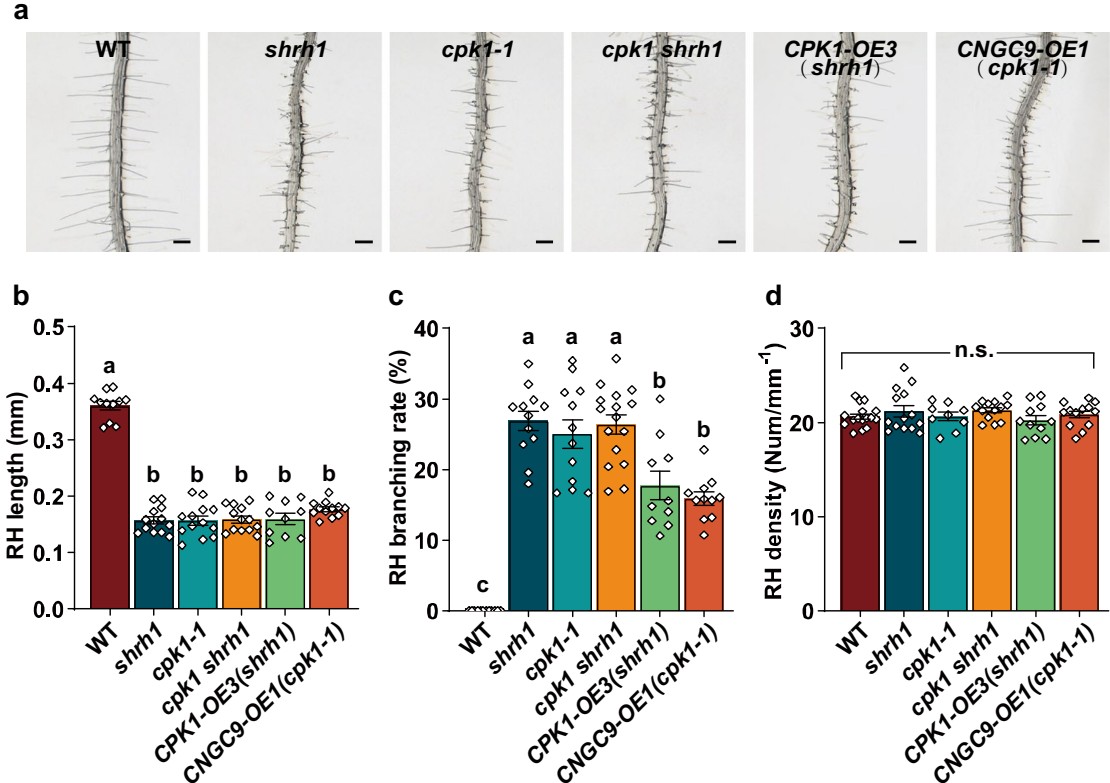

**Fig. 2 | CPK1 functions upstream of CNGC5/6/9 for RH growth.** A set of optical sectioning images of the RHZ of the 4-5 day-old seedlings with a 22.5 μm sectioning step and a sectioning speed of 5 steps per sec were captured under a stereo microscope at room temperature (25 ± 1 °C), and the images were automatically and immediately merged into a 2-D picture for each Petri dish. The merged 2-D pictures were used for RH phenotype analysis. **a** Typical merged 2-D photos of RHZs of individual seedlings. Scale bars, 0.2 mm in (**a**). **b–d** Statistical analyses of RH length (**b**), branching rates (**c**), and RH density (**d**). **b–d** Numbers of biologically independent roots with approximate 50 RHs per root examined are 11, 13, 13, 13, 10, and 12 for RH length (**b**), 12, 12, 12, 16, 10, and 11 for RH branching (**c**), and 16, 14, 9, 13, 11, and 13 for RH density (**d**), for the lines WT, *shrh1*, *cpk1-1*, *cpk1 shrh1*, *CPK1-OE3(shrh1)*, and *CNGC9-OE1(cpk1-1)*, respectively. Samples with different letters are significantly different with $P < 0.05$ (one-way ANOVA), and data are presented as means ± SEM in (**b–d**). Source data are provided as a Source Data file.

*CPK1* in *CPK1-OE3* (*shrh1*) (Supplementary Fig. 2a) and *CNGC9* in *CNGC9-OE1* (*cpk1-1*) (Supplementary Fig. 2d). Analysis of the RH phenotypes of the two overexpression lines revealed shorter RHs in both *CPK1-OE3* (*shrh1*) and *CNGC9-OE1* (*cpk1-1*), similar to the *cpk1-1* and *shrh1* mutants (Fig. 2a, b). In addition, a strong RH branching phenotype was seen in *CPK1-OE3(shrh1)* and *CNGC9-OE1(cpk1-1)* compared to the wild type, and the RH branching phenotypes in the two transgenic lines were weaker than that of *shrh1*, *cpk1-1*, and *cpk1 shrh1* (Fig. 2c). The RH density was not obviously altered in the two overexpression lines compared to the wild type (Fig. 2d). The failure to rescue *cpk1-1* RH phenotypes and to induce longer RHs in the same mutant by the overexpression of *CNGC9* is consistent with CPK1 being an essential upstream activator of CNGC9. Similarly, the failure to rescue RH phenotypes and to induce longer RHs by the overexpression of *CPK1* in *shrh1* is consistent with CNGC5/6/9 being the downstream targets of CPK1. Thus, together the results strongly support our hypothesis that

CPK1 functions upstream of CNGC5/6/9 in the regulation of RH growth in Arabidopsis.

## RH phenotypes of *cpk1* mutants can be partially rescued by high external $Ca^{2+}$

To test the hypothesis that CPK1 regulates RH growth by triggering $Ca^{2+}$ influx through the direct regulation of CNGC5/6/9, we analyzed the effects of external $Ca^{2+}$ on the RH phenotypes of *cpk1-1* and *cpk1-2*. The elongating growth of RHs was strongly promoted by $Ca^{2+}$ in a dose-dependent manner in all the lines, including the wild type, *cpk1-1*, *cpk1-2*, and *CPK1-OE13* (Supplementary Fig. 5a, b). RH elongation was promoted to a lesser extent in the *cpk1-1* and *cpk1-2* mutants compared to that of wild type and *CPK1-OE13*, especially under normal $Ca^{2+}$ level (1.5 mM; Supplementary Fig. 5a, b). The strong RH branching phenotype in *cpk1-1* and *cpk1-2* was strongly suppressed in both low (0 and 0.1 mM) and high (10 mM) $Ca^{2+}$ medium compared to normal levels,

and the branching RH was very rarely observed in the wild type and *CPK1-OE13* plants regardless of the $Ca^{2+}$ level (Supplementary Fig. 5c). RH density was not obviously altered by changes in the $Ca^{2+}$ concentration in any of the Arabidopsis lines tested (Supplementary Fig. 5a, d). These data together suggest that CPK1 functions in RH growth in a $Ca^{2+}$-dependent manner, supporting our hypothesis that CPK1 is the direct upstream regulator of CNGCs and external $Ca^{2+}$ influx in RHs.

## CPK1 physically interacts with CNGC5/6/9

It is well-established that CNGC5/6/9 function as the main components of the plasma membrane (PM) $Ca^{2+}$ channels essential for the regulation of cytosolic $Ca^{2+}$ signaling and RH growth in Arabidopsis[48,49]. To test the possibility that CPK1 functions as the direct upstream regulator of CNGC5/6/9, we analyzed the protein-protein interactions between CPK1 and CNGC5/6/9 using the yeast 2-hybrid (Y2-H) technique based on a mating split-ubiquitin system (mbSUS)[80]. We found that CPK1 interacts strongly with full-length CNGC5/6/9 (Fig. 3a). We then analyzed the interactions of CPK1 with CNGC5/6/9 using split luciferase system in *N. Benthamiana* leaves, and WRKY72 was used as a negative control. We observed clear luciferase luminescent signal in the leaves co-expressing CPK1-nLUC and cLUC-CNGCs, but not in the leaves co-expressing WRKY72-nLUC and cLUC-CNGCs (Fig. 3b). We conducted immuno-blot assays in *N. benthamiana* leaves after the luciferase luminescence detection, and the expression of full length CPK1, WRKY72, and CNGC5/6/9 was detected (Fig. 3c), supporting our luciferase data. We also conducted a bimolecular fluorescence complimentary (BiFC) assay using split YFC in *Xenopus laevis* oocytes as previously described[73,78]. The C terminus ($YFP^C$) and N terminus ($YFP^N$) of YFP were fused to the C termini of CPK1 and CNGC5/6/9, respectively. The fused proteins were designated as CPK1-$YFP^C$, CNGC5-$YFP^N$, CNGC6-$YFP^N$, and CNGC9-$YFP^N$, respectively. cRNA was prepared in vitro, and was micro-injected into the Xenopus oocytes. After a 2-3-day incubation at 16 °C to allow transient gene expression, we observed clear YFP signal in the periphery of the Xenopus oocytes co-injected with the cRNA of CPK1-$YFP^C$ and one of the CNGC5-$YFP^N$, CNGC6-$YFP^N$, and CNGC9-$YFP^N$, but failed to observe any YFP signal in the Xenopus oocytes co-injected with the cRNA of CPK1-$YFP^C$ and the free $YFP^N$ (Fig. 3d), supporting the interaction of CPK1 with CNGC5/6/9, and suggesting that this occurs in the PM. These data together demonstrate clearly that CPK1 physically interacts with CNGC5/6/9.

We further analyzed the interactions of CPK1 to the fragments of CNGC5/6/9 through classical GAL4 Y2-H assay[81], and observed strong interactions between CPK1 and the N termini of CNGC5/6/9 (CNGC5-N, CNGC6-N, and CNGC9-N) (Fig. 4a). We failed to observe obvious interaction of CPK1 with the C termini and transmembrane (TM) domains of the three CNGCs (Fig. 4a). It has been reported that CNGC14 is involved in RH growth for touch sensing[68], but the triple mutant *shrh1* and quadruple mutant *shrh2* showed similar RH phenotypes[48], suggesting that CNGC14 is not essential for RH growth. We thus conducted an Y2-H assay to analyze the potential interactions of CPK1 with fragments of CNGC14. No interaction was detected between CPK1 and the N- or C-terminus or TM domain of CNGC14 (Fig. 4a), implying that CNGC14 isn't a downstream target of CPK1.

We then conducted a luciferase complementation assay in *N. benthamiana* leaves to test for potential interactions between CPK1 and the N termini of CNGC5/6/9, and WRKY72 was used as a negative control. Strong protein interactions of CPK1 with the N termini of CNGC5/6/9 were observed (Fig. 4b), and no obvious protein interaction of WRKY72 with the N termini of CNGC5/6/9 was detected (Fig. 4b). We conducted immuno-blot assays after the luciferase luminescence detection in *N. benthamiana* leaves, which confirmed the successful expression of CPK1, WRKY72, and the N termini of CNGC5/6/9 (Fig. 4c), supporting our luciferase complementation results.

We also performed a co-immunoprecipitation (Co-IP) assay in *N. benthamiana* leaves, which further confirmed the interaction of CPK1 with CNGC5-N (Fig. 4d), CNGC6-N (Fig. 4e), and CNGC9-N (Fig. 4f). All those data together demonstrate that CPK1 physically interacts with the N termini of CNGC5/6/9, supporting our hypothesis that CPK1 is a direct upstream regulator of CNGC5/6/9 for RH growth.

## CPK1 strongly activates CNGC5/6/9

To test whether CPK1 is capable of directly activating CNGC5/6/9, we transiently expressed *CNGC9* and *CPK1* in *Xenopus* oocytes, and measured the whole-oocyte currents using the two-electrode voltage-clamping (TEVC) technique in 30 mM $Ba^{2+}$-based bath solution as described previously[73]. The $Ba^{2+}$-based bath solution is more suitable for divalent cation channel current recordings than a $Ca^{2+}$-based bath solution because $Ca^{2+}$-influx-triggered endogenous anion channel currents can be almost completely abolished in the $Ba^{2+}$-based bath solution in *Xenopus* oocytes[73]. We observed small background whole-oocyte currents in the oocytes injected with $H_2O$ or expressing *CPK1* alone, a modest activation of whole-oocyte currents in oocytes expressing *CNGC9* alone, and a much stronger activation of whole-oocyte currents in the oocytes co-expressing *CNGC9* and *CPK1* (Supplementary Fig. 6a, b). We also tested the effects of inactivated CPK1 with a K179R point mutation (designated as $CPK1^{K179R}$). We observed modest whole-oocyte currents in the oocytes coexpressing $CPK1^{K179R}$ and *CNGC9*, and the activation of CNGC9 was similar to that in the oocytes expressing *CNGC9* alone, but significantly smaller than that in the oocytes co-expressing wild type *CPK1* and *CNGC9* (Supplementary Fig. 6a, b). The CNGC9-mediated whole-oocyte currents were strongly inhibited by the $Ca^{2+}$ channel blocker $Gd^{3+}$ (100 μM) (Supplementary Fig. 6a, b). These data together indicate that CPK1 can activate CNGC9.

We next analyzed the activation of CNGC5/6/9 by CPK1 in human embryonic kidney 293 T (HEK293T) cells which lack the $Ca^{2+}$-influx-activated endogenous anion channel currents. We first performed cytosolic $Ca^{2+}$ imaging experiments by monitoring the FRET/CFP ratio of the $Ca^{2+}$ indicator Cameleon version 3.6 (YC3.6) in HEK293T cells as described[82]. We observed a small cytosolic $Ca^{2+}$ increase in control HEK293T cells, a modest cytosolic $Ca^{2+}$ increase in HEK293T cells expressing wild-type *CNGC5* alone, and a larger cytosolic $Ca^{2+}$ increase in the HEK293T cells co-expressing *CPK1* and *CNGC5*, upon the application of 10 mM external $Ca^{2+}$ (Fig. 5a), demonstrating the activation of CNGC5 by CPK1. Similarly we also observed the activation of CNGC6 (Fig. 5b) and CNGC9 (Fig. 5c) by CPK1 by monitoring the cytosolic $Ca^{2+}$ elevation in HEK293T cells. We tested the activation of CNGC9 by $CPK1^{D274A}$ which is an inactivated version of CPK1 with a D274A point mutation and observed a modest cytosolic $Ca^{2+}$ increase in the HEK293T cells co-expressing CNGC9 and $CPK1^{D274A}$, similar to that seen in the HEK239T cells expressing CNGC9 alone (Fig. 5c), suggesting that CNGC9 was not activated by $CPK1^{D274A}$. These data together demonstrate the activation of CNGC5/6/9 by CPK1.

We next analyzed the activation of CNGC5/6/9 by CPK1 using patch clamping technique by measuring whole-cell $Ca^{2+}$ channel currents in HEK293T cells. We observed small background currents in the mock control HEK293T cells, modest whole cell currents in HEK293T cells expressing CNGC5 alone, and a stronger activation of whole-cell currents in HEK293T cells co-expressing CNGC5 and CPK1 (Fig. 5d, e), demonstrating the activation of CNGC5 by CPK1. Similar activation of CNGC6 (Fig. 5f, g) and CNGC9 (Fig. 5h, i) by CPK1 was observed in the patch clamping analysis in HEK293T cells. We also found that the inactivated $CPK1^{D274A}$ did not activate CNGC9 in HEK293T cells (Fig. 5h, i). These patch clamping data are consistent with the cytosolic $Ca^{2+}$ imaging data and verify the activation of CNGC5/6/9 by CPK1.

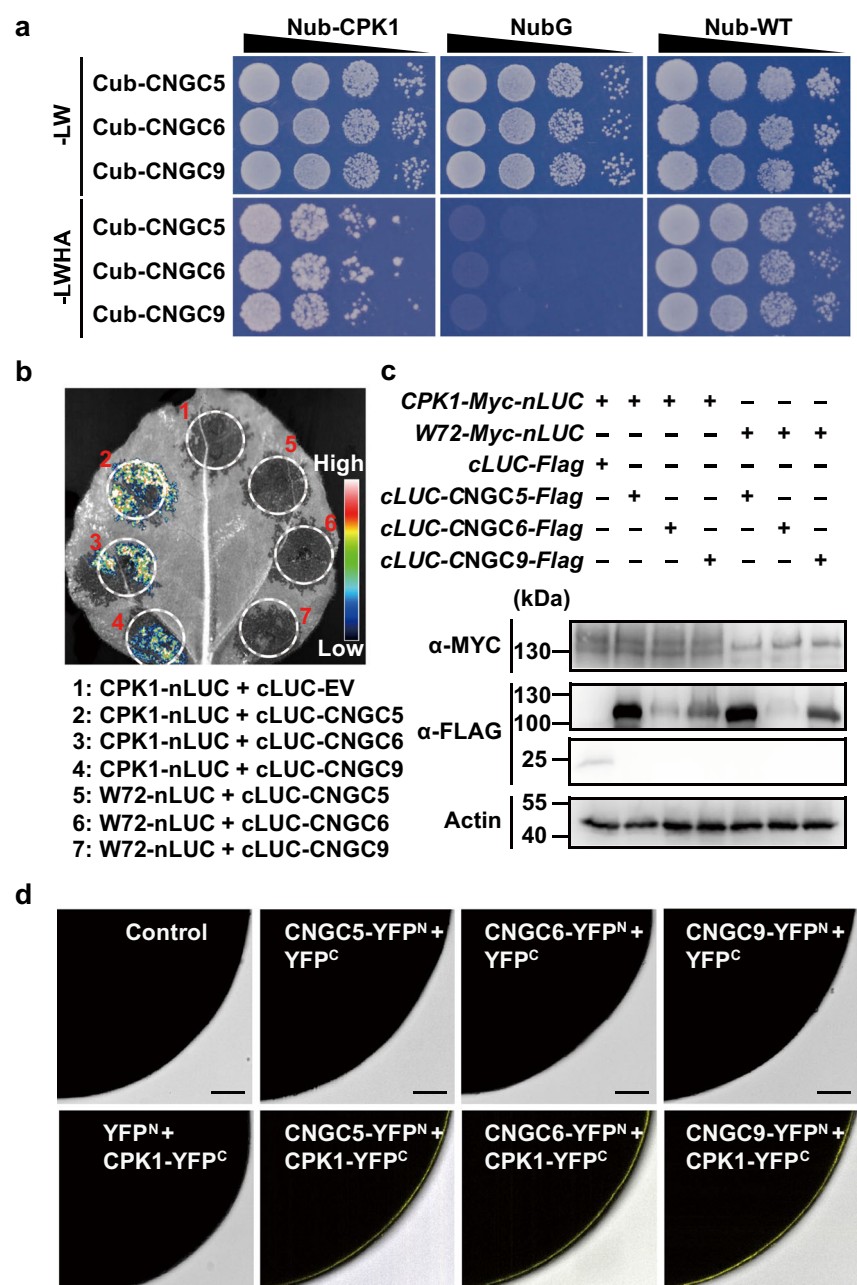

**Fig. 3 | CPK1 physically interacts with CNGC5/6/9. a** Y-2H results show the interactions of CPK1 with CNGC5/6/9. **b** Luciferase complementation analysis in *N. benthamiana* leaves show the protein interactions of CPK1 with CNGC5/6/9, but no interaction of WRKY72 with CNGC5/6/9. **c** Immuno-blot assay data show the expression of *CPK1*, *WRKY72*, and *CNGC5/6/9* in *N. benthamiana* leaves, supporting the luciferase complementation results. **d** BiFC assays in *Xenopus laevis* oocytes show the interactions of CPK1 with CNGC5/6/9 and the subcellular localization of CNGC5/6/9 in the plasma membrane of oocytes. The Y-2H assay was performed using the mbSUS system, and the CDS of *CPK1* and *CNGC5/6/9* were cloned into the vectors *pNXgate32-3HA* (*Nub*) and *pMetYCgate* (*Cub*), respectively. For luciferase complementation assay in *N. benthamiana leaves*, the CDS of *CPK1* was cloned into the modified vector *pCambia1300-nLUC*, the CDS of *CNGC5/6/9* were cloned into the modified vector *pCambia1300-cLUC*. *Agrobacterium* (strain *GV3101*) carrying

the nLUC and cLUC vectors were co-injected into *N. benthamiana* leaves. The leaves were sprayed with 1 mM/L luciferin in the dark 48 hours after the injection, and photos were taken 7 min later. For the BiFC assay in *Xenopus laevis* oocytes, the CDS of *CPK1* and *CNGC5/6/9* were cloned into the vectors *pGEMKN-YFP^C* and *pGEMKN-YFP^N*, respectively. cRNA was transcribed in vitro. A total of volume of 50 nL of the cRNA mixture containing 25 ng of each vector was micro-injected into the oocytes. The oocytes were incubated in the ND96 solution for 2-3 days at $16 \pm 0.5$ °C, and images of the oocytes were captured under a confocal microscope. Three biological replicates were performed for Y2-H, Luciferase complementation, and immune-blot assays. 5 oocytes were analyzed in BiFC assay for each experiment in (**d**). Scale bars, 0.1 mm (**d**). W72 denotes WRKY72 in (**b**) and (**c**). Source data are provided as a Source Data file.

## CPK1 phosphorylates CNGC5/6/9 at a conserved serine site within their N termini

Our data show that CPK1 can interact with and activate CNGC5/6/9 (Figs. 4, 5). We thus hypothesized that CPK1 phosphorylation of these CNGCs may regulate their $Ca^{2+}$ channel activity. To identify the CPK1-

target sites in CNGC9, we analyzed its N-terminal region using mass spectrometry, and 11 candidate sites were identified (Fig. 6a, b; Supplementary Fig. 7a). It has been established that the sites targeted by protein kinases from the CPK and SnRK2 families mainly reside in RxxS motifs[83,84]. Of the 11 candidate sites identified, only Ser26 and Ser73

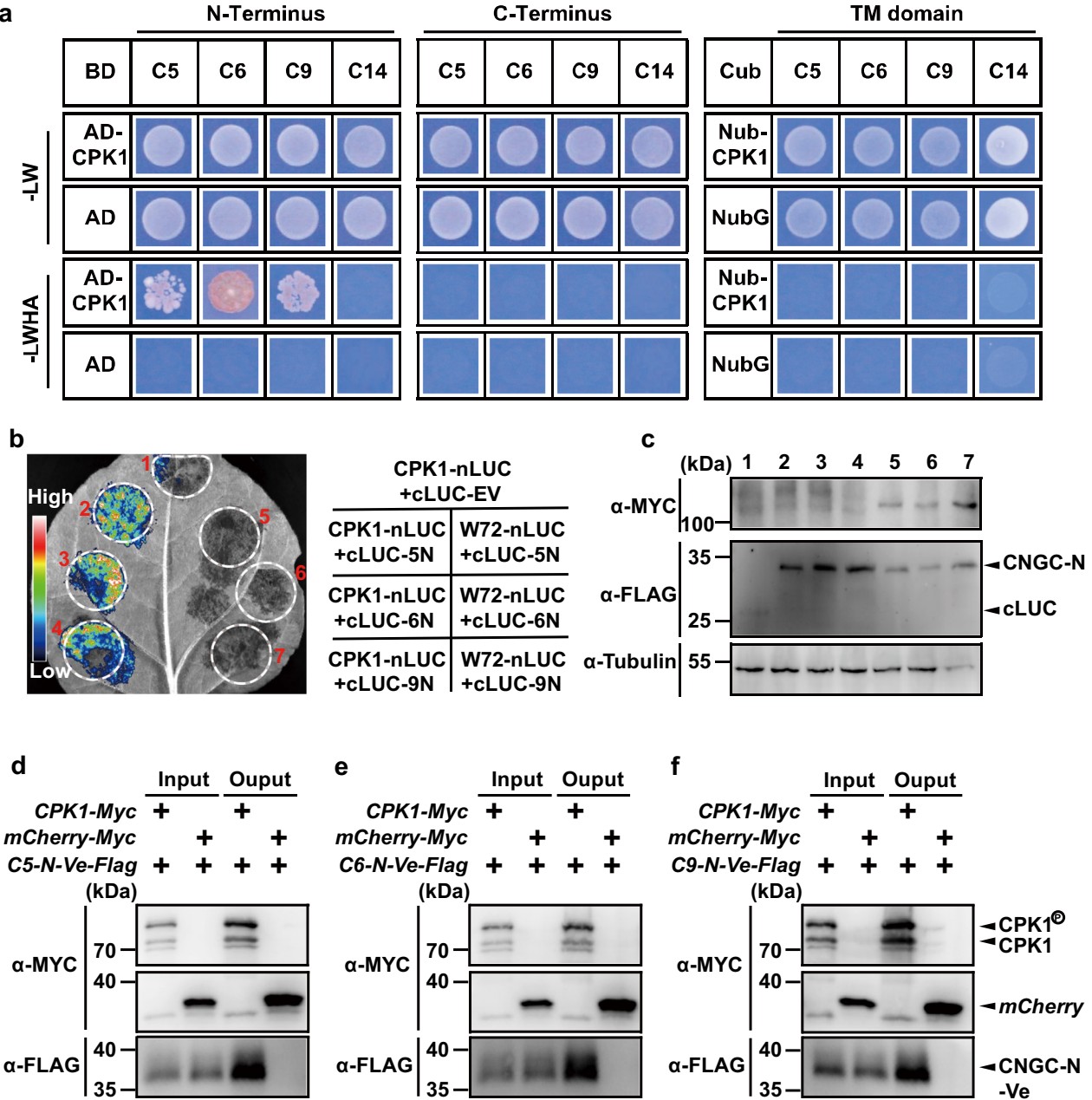

**Fig. 4 | CPK1 interacts with the N termini of CNGC5/6/9. a, b** The interactions of CPK1 with the N termini of CNGC5/6/9 were observed using the Y2-H assay (**a**) and firefly luciferase complementation assay in *N. benthamiana* leaves (**b**). The expression of CPK1 and the N termini of CNGC5/6/9 in *N. benthamiana* leaves was verified by immuno-blot assay (**c**). The interactions of CPK1 to the N termini of CNGC5/6/9 were further verified by in vitro Co-IP: CNGC5-N (**d**), CNGC6-N (**e**), and CNGC9-N (**f**). The Y2-H assay was performed using the classical GAL4 Y2-H system. For the in vitro Co-IP assay, the CDS of *CPK1* and *CNGC5-N, CNGC6-N, CNGC9-N* were cloned into the vectors *1305-UBQ10: Myc* and *1305-UBQ10-Venus-Flag*, respectively, downstream the *UBQ10* promoter. The vectors were co-transformed into *N. benthamiana* leaves, and the combined expression of *mCherry-Myc* and *CNGC9-N-Venus-Flag* was used as control. The fused proteins were extracted, and the antibodies against the Myc and Flag tags were used for immuno-blot analysis. Three biological replicates were conducted for all experiments. Abbreviations: C5 (CNGC5), C6 (CNGC6), C9 (CNGC9), C14 (CNGC14), and TM (transmembrane) in (**a**), W72 (WRKY72), 5 N (CNGC5-N), 6 N (CNGC6-N), and 9 N (CNGC9-N) in (**b**), C5-N-Ve (CNGC5-N-Venus) in (**d**), C6-N-Ve (CNGC6-N-Venus) in (**e**), and C9-N-Ve (CNGC9-N-Venus) in (**f**). Source data are provided as a Source Data file.

reside in an RxxS motif (Supplementary Fig. 7a), and Ser26 was previously identified as the main SnRK2.6/OST1-target site[73]. We then generated point-mutated versions of CNGC9-N by substituting the candidate serine residues with Ala (A) to prevent the phosphorylation of these sites (CNGC9-N$^{S26A}$ and CNGC9-N$^{S73A}$). We expressed CPK1, CNGC9-N$^{S26A}$ and CNGC9-N$^{S73A}$ in the *E. coli* strain *Rosetta-gami(DE3) pLysS*, and purified the proteins as previously described[85]. We then performed in vitro protein phosphorylation assays in the presence of 0.5 mM CaCl$_2$, and found that both wild-type CNGC9-N and CNGC9-

N$^{S73A}$ were strongly phosphorylated by CPK1, but the phosphorylation of CNGC9-N$^{S26A}$ by CPK1 was dramatically attenuated (Fig. 6c). These results suggest that the Ser26 is the main CPK1-target site on CNGC9-N, and Ser73 may be a secondary CPK1-target site.

We then analyzed the Ca$^{2+}$ dependence of CPK1-mediated protein phosphorylation. Autophosphorylation of CPK1 was observed in the presence of 0.5 mM CaCl$_2$, but was strongly inhibited in Ca$^{2+}$-free reaction buffer supplemented with 5 mM EGTA (Fig. 6c). Moreover, we found that CPK1 phosphorylation of CNGC9-N was slightly weaker in

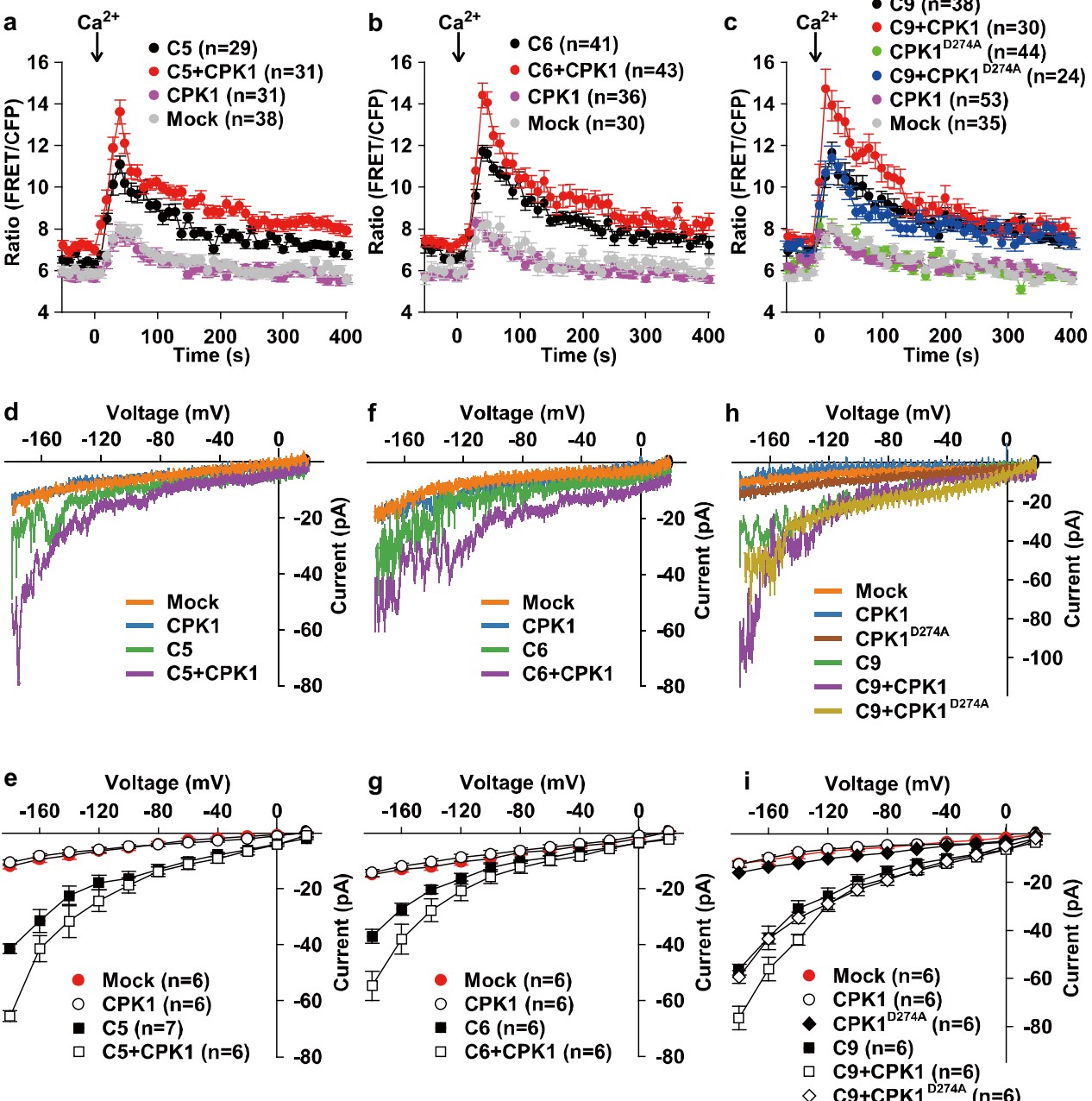

**Fig. 5 | CPK1 activates CNGC5/6/9 in HEK293T cells.** Cytosolic Ca²⁺ imaging and patch clamping analysis were performed in HEK293T cells, and the activation of CNGC5/6/9 by CPK1, but not by CPK1$^{D274A}$, was observed. CPK1, CPK1$^{D274A}$, and CNGC5/6/9 were cloned into the vector *pCI-neo-IRES-YC3.6*, and the vectors were then transformed into HEK293T cells using the Lipofectamine 2000 Transfection Reagent Kit individually or in a combination as indicated. HEK293T cells were cultured for 2-3 days after transformation, and the HEK293T cells with bright YC3.6 fluorescent signal were used for cytosolic Ca²⁺ imaging and patch clamping analysis. **a**–**c** Cytosolic Ca²⁺ imaging data show the activation of CNGC5 (**a**), CNGC6 (**b**), and CNGC9 (**c**) by CPK1, but not by CPK1$^{D274A}$ (**c**). 10 mM external Ca²⁺ was added as indicated by the arrows. **d**–**i** Typical whole-cell recordings (**d, f,** and **h**) and average current-voltage curves (**e, g,** and **i**) of patch clamping results show the activation of CNGC5 (**d** and **e**), CNGC6 (**f** and **g**), and CNGC9 (**h** and **i**) by CPK1. C5, C6, and C9 represent CNGC5, CNGC6, and CNGC9, respectively. The numbers of HEK293T cells tested are indicated in brackets, and data are presented as means ± SEM in (**a**–**c**), (**e**), (**g**), and (**i**). Source data are provided as a Source Data file.

the absence of Ca²⁺ (Fig. 6c). This indicates that the autophosphorylation of CPK1 is strongly dependent on Ca²⁺, but the phosphorylation of CNGC9-N by CPK1 is largely independent of the Ca²⁺-dependent autophosphorylation of CPK1.

We conducted an alignment of the amino sequences among the N termini of CNGC5/6/9, which showed that Ser26 in CNGC9 corresponds to Ser20 in CNGC5 and Ser27 in CNGC6, which in both cases similarly reside in an RxxS motif (Supplementary Fig. 7b). The other CPK1-target site, S73 in CNGC9-N, was found to correspond to Ser58

in CNGC5 and Ser73 in CNGC6 (Supplementary Fig. 7b). Note, Ser73-CNGC6 and Ser73-CNGC9 reside within an RxxS motif, and Ser58-CNGC5 does not reside within an RxxS motif (Supplementary Fig. 7b), suggesting that these three serines are secondary CPK1-target sites that may not be essential for the activation of CNGC5/6/9. Thus, the results from our protein phosphorylation assays (Fig. 6c) suggest that the main CPK1-target site in RH-expressed CNGCs is conserved, corresponding to Ser20 in CNGC5, Ser27 in CNGC6, and Ser26 in CNGC9.

## Phosphorylation of CNGC5/6/9 at the main CPK1-target sites is required for the activation of CNGC5/6/9

To study whether the phosphorylation CNGC5/6/9 at their main CPK1-target sites is required for their activity as $Ca^{2+}$ channels, we substituted their main CPK1-target Ser (S) sites with Ala (A) to create phospho-dead versions. The point-mutated versions of CNGC5/6/9 were designated as $CNGC5^{S20A}$, $CNGC6^{S27A}$, and $CNGC9^{S26A}$. We transiently co-expressed CPK1 with either wild-type or phospho-dead CNGC5/6/9, or each one individually, in HEK293T cells, and performed cytosolic $Ca^{2+}$ imaging experiments. We observed a small cytosolic $Ca^{2+}$ increase in the mock control HEK293T cells and in the HEK293T cells expressing either CPK1 or $CNGC5^{S20A}$ alone, a modest cytosolic $Ca^{2+}$ increase in the HEK293T cells expressing wild-type CNGC5 alone, and a much larger cytosolic $Ca^{2+}$ increase in the HEK293T cells co-expressing CPK1 and CNGC5, upon the application of 10 mM external $Ca^{2+}$ (Fig. 7a). The cytosolic $Ca^{2+}$ increase in the HEK293T cells expressing $CNGC5^{S20A}$ alone was similar to that in the HEK293T cells co-expressing CPK1 and $CNGC5^{S20A}$ (Fig. 7a), demonstrating that the S20A point-mutation dramatically impaired the activation of CNGC5 by CPK1. These data together established that the phosphorylation of CNGC5 by CPK1 at Ser20 is necessary for the activation of CNGC5 by CPK1. In other words, Ser20 is the main CPK1-target site for the activation of CNGC5. Similar cytosolic $Ca^{2+}$ imaging results were also obtained for CNGC6 (Fig. 7b) and CNGC9 (Fig. 7c) in HEK293T cells. These results together establish that S20, S27, and S26 are the main CPK1-target sites for the respective activation of CNGC5/6/9.

We next performed patch clamping experiments to analyze the activation of CNGC5/6/9 by CPK1 by measuring whole-cell $Ca^{2+}$ channel currents in HEK293T cells in a 10 mM $Ca^{2+}$-based bath solution as previously described[73]. We observed small background conductance in the mock control HEK293T cells and the HEK293T cells expressing CPK1 alone. Similarly, a small activation of whole-cell currents was seen in cells expressing $CNGC5^{S20A}$ alone or coexpressing $CNGC5^{S20A}$ and CPK1. By contrast, a modest activation of whole-cell $Ca^{2+}$ channel currents was seen in cells expressing wild-type CNGC5, and much larger whole-cell $Ca^{2+}$ channel currents were seen in cells co-expressing wild-type CPK1 and CNGC5 (Fig. 7d, e). The whole-cell currents in HEK293T cells expressing $CNGC5^{S20A}$ alone were similar to those in HEK293T cells coexpressing $CNGC5^{S20A}$ and CPK1, but were significantly smaller than those of cells either expressing wild-type CNGC5 or co-expressing CNGC5 and CPK1 (Fig. 7d, e). Similar patch clamping results were also observed in HEK293T cells for the CPK1-activation of CNGC6 (Fig. 7f, g) and CNGC9 (Fig. 7h, i). These data are consistent with our cytosolic $Ca^{2+}$ imaging results in HEK293T cells, and further demonstrate that the phosphorylation of CNGC5/6/9 by CPK1 is necessary for the activation of the three CNGCs.

## The phosphorylation of CNGC5/6/9 at the main CPK1-target sites is necessary and sufficient to rescue the RH phenotypes of the *shrh1* triple mutant and *cpk1-1*

It has been reported that OST1 phosphorylation of CNGC5/6/9/12 at the main OST1-target sites is required for ABA-induced stomatal closure[73], and the previously identified OST1-target sites in CNGC5/6/9 are identical to the CPK1-target sites identified in this study. Phospho-dead versions of CNGC5/6/9/12, including $CNGC5^{S20A}$, $CNGC6^{S27A}$, $CNGC9^{S26A}$, and $CNGC12^{S13A}$, largely lack $Ca^{2+}$ channel activity, were designated as iCNGCs (inactivated CNGCs)[73]. Conversely, the phospho-mimic versions of CNGC5/6/9/12, including $CNGC5^{S20D}$, $CNGC6^{S27D}$, $CNGC9^{S26D}$, and $CNGC12^{S13D}$, showed a strong activity that was similar to their activation by OST1[73], so these were designated as aCNGCs (activated CNGCs)[73]. Hereafter we will use the same iCNGC and aCNGC designations to refer to these mutants.

It has been well-established that CNGCs are capable of forming homo- and hetero-tetramers[71,72], and that CNGC5/6/9 are important for RH growth in Arabidopsis[48]. We thus hypothesized that RH phenotypes

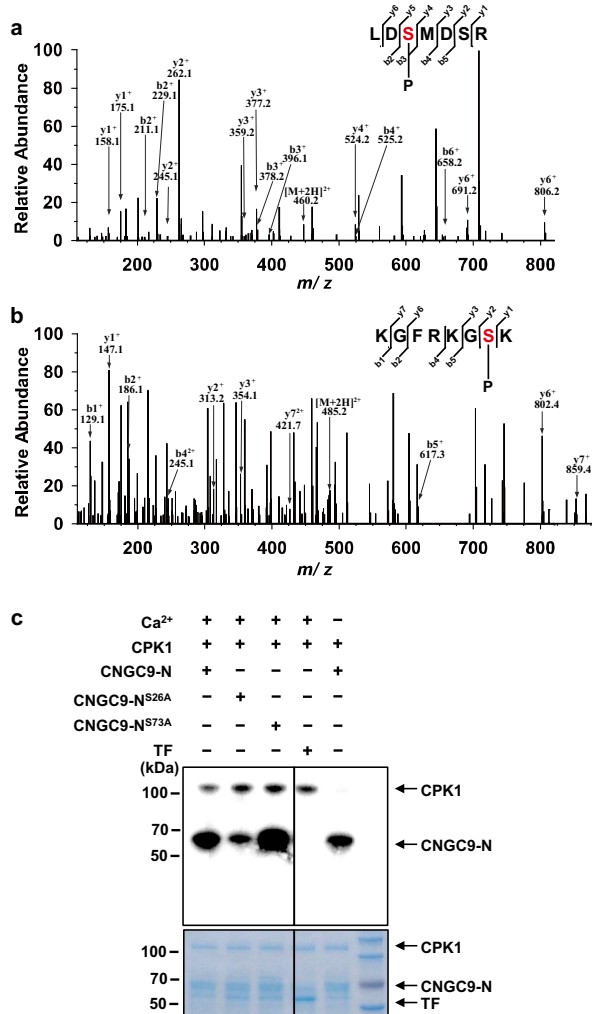

**Fig. 6 | The main target site of CPK1 at the N terminus of CNGC9 is Ser26. a, b** LC-MS/MS spectrum show that the peptides LDSMDSR and KGFRKGSK contain the phospho-serines Ser26 (**a**) and Ser73 (**b**). **c** In vitro phosphorylation assay shows that the CPK1-mediated phosphorylation of $CNGC9$-$N^{S26A}$ was dramatically reduced, whereas the phosphorylation of $CNGC9$-$N^{S73A}$ was not obviously attenuated, compared to wild-type CNGC9-N. For the in vitro phosphorylation assays, the wild type and point-mutated CNGC9-N were cloned into the vector *pCold-TF* with a cleavable 6×His tag, and CPK1 was cloned into the vector *pMAL-c5X* with a MBP tag. The recombinants were expressed in *E. coli* and purified, and the recombinants with the tags retained were used for the in vitro protein phosphorylation assays. Three biological replicates were conducted for both the LC-MS/MS and the in vitro phosphorylation assays. Source data are provided as a Source Data file.

of *shrh1* may be caused by the lack of CPK1-activated CNGCs. To investigate this, we generated transgenic Arabidopsis lines expressing the *iCNGCs* and *aCNGCs* under their respective native promoters in the background of *shrh1*, and designated these lines as $CNGC5^{S20A}$(*shrh1*) (*expressing $CNGC5^{S20A}$ under CNGC5 native promoter in the background of shrh1*), $CNGC5^{S20D}$(*shrh1*), $CNGC6^{S27A}$(*shrh1*), $CNGC6^{S27D}$(*shrh1*), $CNGC9^{S26A}$(*shrh1*), and $CNGC9^{S26D}$(*shrh1*). The transgenic expression of these CNGC variants was verified by RT-qPCR (Supplementary Fig. 2b–d), and the RH phenotypes were evaluated. We found that the short RH phenotype of *shrh1* was successfully rescued by the expression of *aCNGCs* in the transgenic lines $CNGC5^{S20D}$(*shrh1*), $CNGC6^{S27D}$(*shrh1*), and $CNGC9^{S26D}$(*shrh1*), and the average RH length of $CNGC9^{S26D}$(*shrh1*) was even longer than that in the wild type

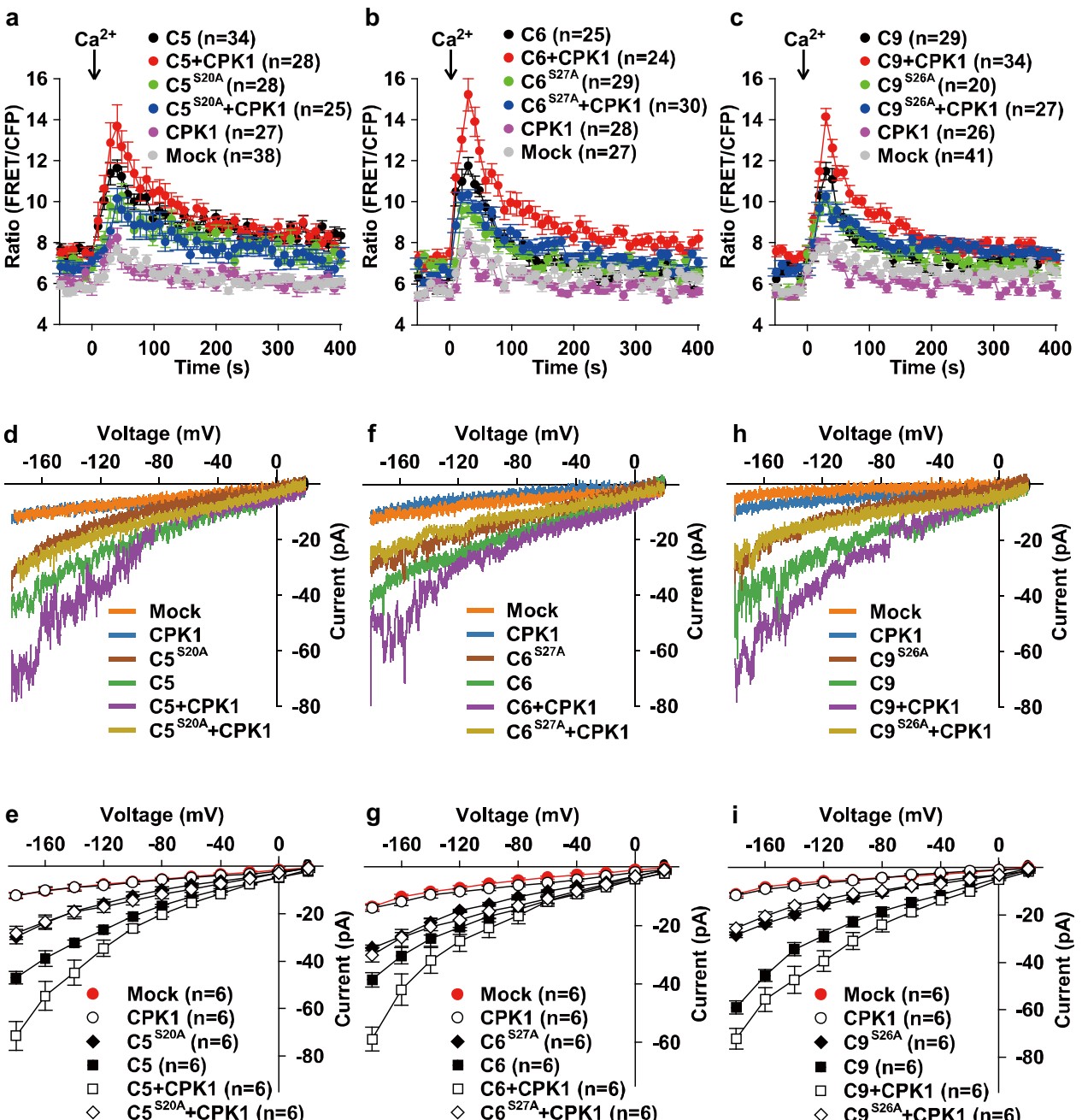

**Fig. 7 | Phosphorylation at the main CPK1-target sites is required for the activation of CNGC5/6/9 in HEK293T cells.** CPK1, the wild type CNGC5/6/9, and iCNGCs with a S-to-A point mutation were cloned into the vector *pCI-neo-IRES-YC3.6*, and were then expressed in HEK293T cells individually or in a combination as indicated. HEK293T cells with bright YC3.6 fluorescent signal were used for cytosolic Ca²⁺ imaging and patch clamping analysis. **a–c** Cytosolic Ca²⁺ imaging data show the strong inhibition of CNGC5 (**a**), CNGC6 (**b**), and CNGC9 (**c**) by the S-to-A point mutations at the main CPK1-target sites compared to the strong activation of wild type CNGCs by CPK1. **d–i** Patch clamping data show the inhibition of CNGC5 (**d**, **e**), CNGC6 (**f**, **g**), and CNGC9 (**h**, **i**) by the S-to-A point mutations at the main CPK1-target sites compared to the activation of wild type CNGC5/6/9 by CPK1. C5^S20A, C6^S27A, and C9^S26A represent CNGC5^S20A, CNGC6^S27A, and CNGC9^S26A, respectively. The numbers of HEK293T cells tested are indicated in brackets, and data are presented as means ± SEM in (**a–c**), (**e**), (**g**), and (**i**). Source data are provided as a Source Data file.

(Fig. 8a, b). Moreover, the RH branching phenotype was also successfully rescued by the expression of *aCNGCs* (Fig. 8a, c). By contrast, we found that the short RHs and RH branching phenotypes were not rescued in the *CNGC5^S20A(shrh1)*, *CNGC6^S27A(shrh1)*, and *CNGC9^S26A(shrh1)* transgenic lines (Fig. 8a–c). Notably, the RH density was not affected in any of the transgenic lines (Fig. 8a, d). This demonstrates that the CPK1 phosphorylation of CNGC5/6/9 is necessary for RH growth in Arabidopsis.

As described above, we hypothesized that the RH phenotypes of *cpk1* mutants are caused by the failure of CPK1 activation of CNGC5/6/9. We then tested whether aCNGCs can rescue the RH growth defects in the *cpk1* mutant. We generated transgenic lines expressing the *iCNGCs* and *aCNGCs* under their respective native promoters in the *cpk1-1* background, and designated these lines as *CNGC5^S20A(cpk1-1)* (*expressing CNGC5^S20A under CNGC5 native promoter in the background of cpk1-1*), *CNGC5^S20D(cpk1-1)*, *CNGC6^S27A(cpk1-1)*,

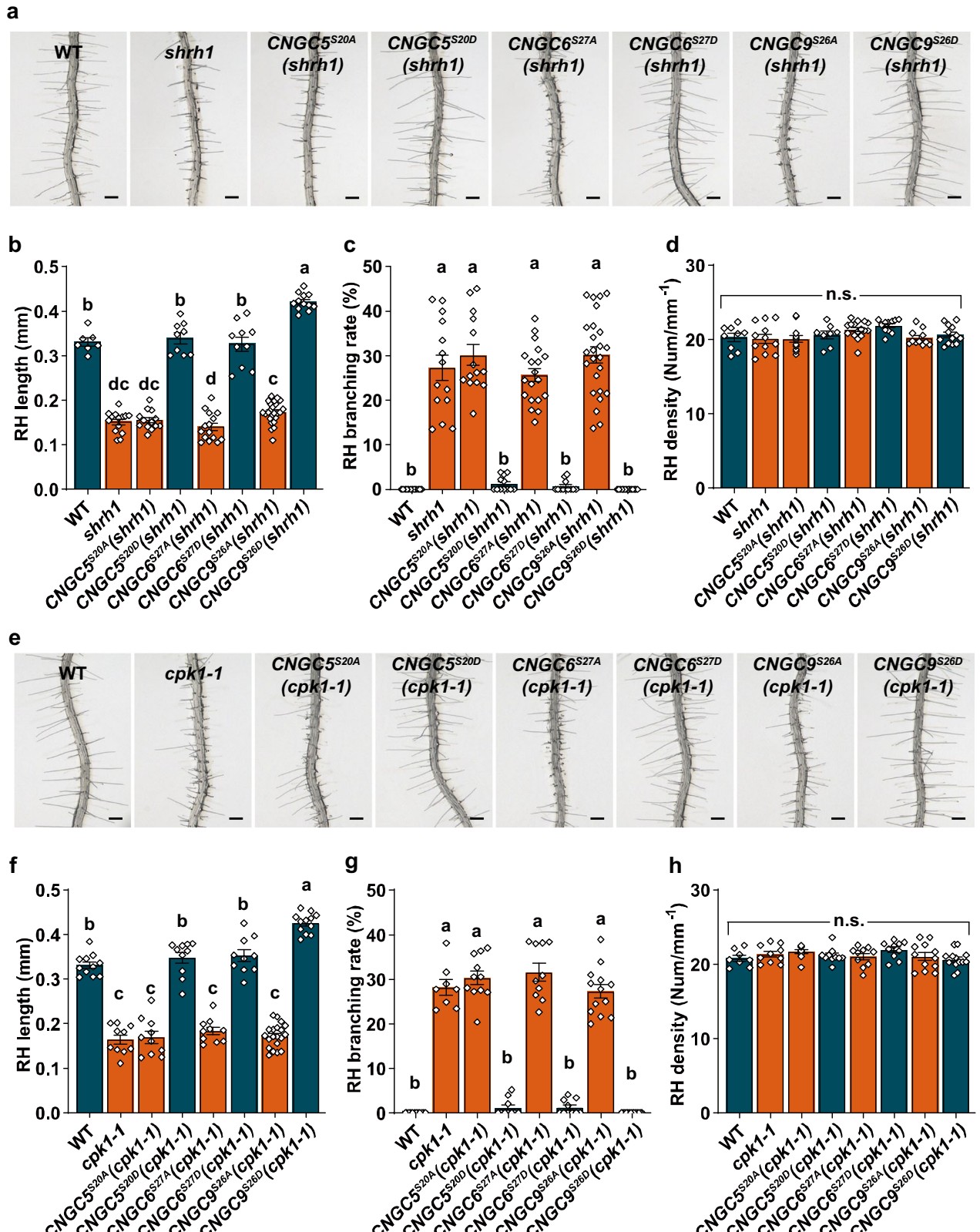

*CNGC9^{S26A}*(*cpk1-1*), and *CNGC9^{S26D}*(*cpk1-1*). The expression of the *iCNGCs* and *aCNGCs* in those transgenic lines was verified by RT-qPCR (Supplementary Fig. 2b–d), and the RH phenotypes were analyzed. This revealed that the expression of the *aCNGCs*, but not the *iCNGCs*, could fully rescue the RH length and branching phenotypes of *cpk1-1* (Fig. 8e–g). Similar to the *shrh1* mutant, the RH density of *cpk1-1* was not significantly altered by the expression of either *iCNGCs* or *aCNGCs*

(Fig. 8e, h). These results indicate that CPK1 phosphorylates CNGC5/6/9 to promote RH growth in Arabidopsis.

We also generated transgenic Arabidopsis lines by overexpressing the *iCNGCs* and *aCNGCs* under the *UBQ10* promoter in the wild type background. The overexpression lines were selected and designated as *CNGC5^{S20A}-OE(WT)* (*OVEREXPRESSING CNGC5^{S20A} under UBQ10 promoter in the wild type background*), *CNGC5^{S20D}-OE(WT)*, *CNGC6^{S27A}-*

**Fig. 8 | The RH phenotypes of *shrh1* and *cpk1-1* can be rescued by aCNGCs, but not by iCNGCs.** A set of optical sectioning images of the RHZ of the 4-5 day-old seedlings with a 22.5-μm sectioning step and a sectioning speed of 5 steps per sec were captured under a stereo microscope at room temperature ($25 \pm 1$ °C), and each set of images were automatically and immediately merged into a 2-D picture for each Petri dish. The merged 2-D pictures were used for RH phenotype analysis. **a–d** Typical merged 2-D photos of the RHZ of the seedlings (**a**), and the statistical analyses of RH length (**b**), RH branching rates (**c**), and RH density (**d**), showing that the RH phenotypes of *shrh1* were rescued by the aCNGCs, but not by the iCNGCs. **b–d** Numbers of biologically independent roots with approximately 50 RHs per root examined are 7, 14, 13, 9, 15, 10, 25, and 12 in (**b**), 12, 14, 14, 12, 19, 12, 25, and 14 in (**c**), and 9, 12, 13, 8, 16, 11, 11, and 13 in (**d**), for the Arabidopsis lines as shown from left to right in each panel. **e–h** Typical merged 2-D photos of the RHZ of the seedlings (**a**) and statistical analyses of RH length (**f**), RH branching rates (**g**), RH density (**h**), showing that the RH phenotypes of *cpk1-1* were rescued by the aCNGCs, but not by the iCNGCs. **e–h** Numbers of biologically independent roots with approximate 50 RHs per root examined are 11, 10, 10, 10, 10, 10, 20, and 12 in (**f**), 12, 8, 11, 10, 10, 10, 13, and 14 in (**g**), and 8, 10, 10, 10, 10, 10, 12, and 13 in (**h**), for the Arabidopsis lines as shown from left to right in each panel. Samples with different letters were found to be significantly different with a $P < 0.05$ (one-way ANOVA), and the data are presented as means ± SEM in (**b–d**) and (**f–h**). Source data are provided as a Source Data file.

---

*OE*(WT), *CNGC6*[S27D]-*OE*(WT), *CNGC9*[S26A]-*OE*(WT), and *CNGC9*[S26D]-*OE*(WT). The overexpression of *aCNGCs* and *iCNGCs* in the transgenic lines was confirmed by RT-qPCR analysis (Supplementary Fig. 2b–d). We analyzed the RH phenotypes, and observed *cpk1*- and *shrh1*-like defects of RH growth, including shorter RHs and more RH branching, in the lines overexpressing the *iCNGCs*, compared to the wild type (Supplementary Fig. 8a–c). This is consistent with a previous study that showed the overexpression of *iCNGCs* strongly suppresses the endogenous wild type CNGCs dominant-negatively[73]. We did not observe any obvious RH length and branching phenotypes in the transgenic lines overexpressing the *aCNGCs* compared to the wild type (Supplementary Fig. 8a–c). RH density was not obviously altered across the transgenic lines expressing either *iCNGCs* or *aCNGCs* (Supplementary Fig. 8a, d). These results support the requirement of CPK1-mediated phosphorylation of CNGCs for RH growth in Arabidopsis.

We then studied the effects of CPK1-mediated phosphorylation at the secondary CPK1-target sites in vivo. We generated point-mutated versions CNGC5[S58A], CNGC5[S58D], CNGC6[S73A], CNGC6[S73D], CNGC9[S73A], and CNGC9[S73D]. We performed cytosolic $Ca^{2+}$ imaging analysis in HEK293T cells and found that the $Ca^{2+}$ channel activity of CNGC9[S73A] was similar to that of wild type CNGC9 (Supplementary Fig. 7c), suggesting that the phosphorylation of that site is not essential for the activation of CNGC9. We then fused an eGFP to the C termini of the mutated versions of CNGC5/6/9 and generated transgenic Arabidopsis lines by expressing these mutated CNGC5/6/9 under their respective native promoters in *cpk1-1*. The lines were designated as *CNGC5*[S58A](*cpk1-1*) (*expressing CNGC5*[S58A] *under CNGC5 native promoter in the background of cpk1-1*), *CNGC5*[S58D](*cpk1-1*), *CNGC6*[S73A](*cpk1-1*), *CNGC6*[S73D](*cpk1-1*), *CNGC9*[S73A](*cpk1-1*), and *CNGC9*[S73D](*cpk1-1*), respectively. RT-qPCR was then used to verify the expression of these transgenes in these lines (Supplementary Fig. 2b–d), and their RH phenotypes were evaluated. We observed *cpk1-1*-like short RHs (Supplementary Fig. 9a, b) and RH branching (Supplementary Fig. 9a, c) phenotypes, and saw no changes in RH density in these transgenic lines (Supplementary Fig. 9a, d), compared to wild type. Those data establish that the secondary CPK1-target sites at the N termini of CNGC5/6/9 are not essential for RH growth in Arabidopsis.

### The phosphorylation of CNGC5/6/9 by CPK1 is required to retain the cytosolic $Ca^{2+}$ gradient at RH tips

To study whether CPK1 regulates RH growth by regulating the cytosolic $Ca^{2+}$ signaling at RH tips via its regulation of CNGC5/6/9, we introduced the gene encoding the calcium indicator YC3.6 under the *UBQ10* promoter in the mutants and transgenic lines described above. To be sure that the results across the lines were comparable, we analyzed their YFP signal fluorescence at RH apices. This revealed that the lines had similar YFP signal intensities (Supplementary Fig. 10), suggesting that the YC3.6 reporter is expressed at similar levels across the lines. We then analyzed the $Ca^{2+}$ gradient at the apex of RHs in these lines. A sharp $Ca^{2+}$ gradient was observed at the RH apex in the wild type plants, but the $Ca^{2+}$ gradients were severely attenuated in the *cpk1-1*, *shrh1*, and *cpk1 shrh1* mutants (Fig. 9a, b). Further analysis revealed that the RH cytosolic $Ca^{2+}$ gradients in the *cpk1-1* and *shrh1*

mutants were successfully restored by the expression of the *aCNGCs*, but not by the *iCNGCs* (Fig. 9a, b).

Overall, our findings indicate that CPK1-mediated phosphorylation of CNGC5/6/9 is necessary and sufficient to maintain the sharp cytosolic $Ca^{2+}$ gradient at RH tips which is required to maintain their growth.

## Discussion

It has been well-established that the cytosolic $Ca^{2+}$ gradient at the apex of RHs plays a central role in RH growth[45,46], and that PM $Ca^{2+}$ channels are essential for RH elongation[86]. Multiple CNGCs have been identified as being essential for RH elongation by our group and others[48,49,68]. In this research, we reveal the requirement of CPK1-mediated phosphorylation of CNGC5/6/9 to establish and maintain the sharp cytosolic $Ca^{2+}$ gradient at the RH apex that drives RH growth. In a word, CPK1 is the main upstream activator of CNGC5/6/9 in RHs. However, we noticed that iCNGCs exhibited a low level of $Ca^{2+}$ channel activity in HEK293T cells in the absence of CPK1 compared with that of wild-type CNGCs (Figs. 5, 7), suggesting that wild-type CNGCs were modestly activated. It is very likely that the three wild-type CNGCs were slightly phosphorylated at the main CPK1-target sites or other yet-to-be-identified sites by endogenous protein kinases in HEK293T cells, and the impairment of CPK1-activation of the three iCNGCs (Fig. 7) supports this explanation.

It has been reported that CNGC14 was expressed in RHs, but the *cngc14* mutant showed a short RH phenotype only when the roots and RHs were embedded in solid medium[68]. Thus, CNGC14 is important for touch sensing for RHs. Our previous study found that the triple mutant *shrh1* (*cngc5/6/9*) and quadruple mutant *shrh2* (*cngc5/6/9/14*) showed quite similar RH phenotypes[48]. In this research we found that CPK1 interacts with CNGC5/6/9, and did not observe obvious protein interaction of CPK1 to CNGC14 in Y2-H assays (Fig. 4a). The negative Y2-H results can't completely exclude the possible protein interaction between CPK1 and CNGC14. Taken together, it is reasonable to conclude that CPK1 is involved in RH growth mainly through activating CNGC5/6/9, the three main CPK1-targets, and CNGC14 may function as a secondary CPK1-target in RHs.

Our data indicate that CPK1-CNGC modules control RH growth by mediating and regulating external $Ca^{2+}$ influx at RH tips. However, the RH phenotypes of the single mutants *cpk1-1* and *cpk1-2* were strongly diminished under both low $Ca^{2+}$ (0 mM and 0.1 mM) and high $Ca^{2+}$ (10 mM) conditions relative to that under the standard (1.5 mM) $Ca^{2+}$ condition (Supplementary Fig. 5a-c). RH elongation was inhibited to a similar extent in wild type, *cpk1* mutants and *OE* lines, suggesting that the amount of $Ca^{2+}$ absorbed by the RHs of wild type and the *CPK1-OE13* transgenic line was reduced to an extent similar to that of *cpk1* under low $Ca^{2+}$. Conversely, the RH growth defects in the *cpk1* mutants were partially rescued by high $Ca^{2+}$, suggesting that over-loading of $Ca^{2+}$ occurred to a similar extent in these lines.

The autophosphorylation of CPK1 showed a strong $Ca^{2+}$-dependence, while the CPK1 phosphorylation of CNGC9-N was less so (Fig. 6c). It is unknown whether CPK1 autophosphorylation is important for RH growth, but it is reasonable to speculate that CPK1 mediates and activates CNGC5/6/9 for cytosolic $Ca^{2+}$ elevation which

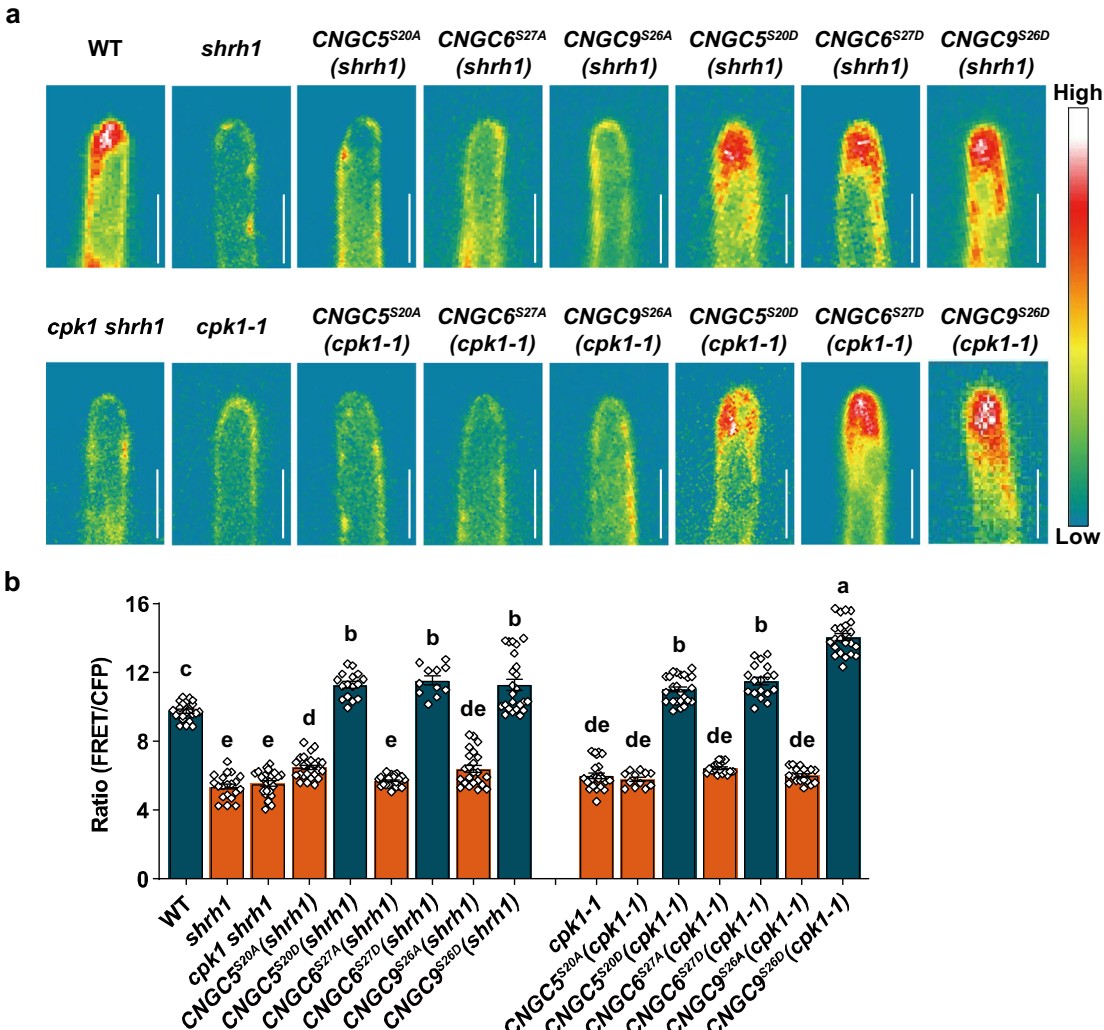

**Fig. 9 | CPK1-mediated phosphorylation of CNGC5/6/9 is necessary and sufficient to retain a sharp cytosolic Ca²⁺ gradient at the apex of elongating RHs.** The cytosolic $Ca^{2+}$ gradient at the tips of growing RH was monitored by measuring the FRET/CFP ratio of YC3.6 in the Arabidopsis lines expressing the calcium sensor YC3.6. A sharp cytosolic $Ca^{2+}$ gradient was observed in wild-type RHs, but this was dramatically attenuated in the *cpk1-1*, *shrh1*, and *cpk1 shrh1* mutants. The sharp cytosolic $Ca^{2+}$ gradient was successfully restored to wild-type-like levels in the RHs of *cpk1-1* and *shrh1* by the expression of *aCNGCs*, but not by the expression of the *iCNGCs*, under their respective native promoters. **a** Pseudo images of typical YC3.6 FRET/CFP ratios in RHs of wild-type, *cpk1-1*, *shrh1*, and *cpk1 shrh1* mutants, and the

*CNGC5^{S20A}(shrh1)*, *CNGC5^{S20D}(shrh1)*, *CNGC6^{S27A}(shrh1)*, *CNGC6^{S27D}(shrh1)*, *CNGC9^{S26A}(shrh1)*, *CNGC9^{S26D}(shrh1)*, *CNGC5^{S20A}(cpk1-1)*, *CNGC5^{S20D}(cpk1-1)*, *CNGC6^{S27A}(cpk1-1)*, *CNGC6^{S27D}(cpk1-1)*, *CNGC9^{S26D}(cpk1-1)*, and *CNGC9^{S26A}(cpk1-1)* transgenic lines. **b** YC3.6 FRET/CFP ratios reflecting cytosolic $Ca^{2+}$ gradients at RH apices. Scale bars, 10 µm in (**a**). $n = 24, 20, 25, 25, 16, 25, 11, 23, 24, 24, 12, 24, 18, 18, 19$, and $23$ biologically independent RHs for the lines as shown from left to right in (**b**). The data are presented as means ± SEM, and samples with different letters are significantly different with $P < 0.05$ (one-way ANOVA) in (**b**). Source data are provided as a Source Data file.

consequently enhances the activity of CPK1 via autophosphorylation to further elevate cytosolic $Ca^{2+}$ levels. Thus, CPK1 and CNGC5/6/9 may operate as a positive feedback loop to maintain the sharp cytosolic $Ca^{2+}$ gradient at RH tips to facilitate RH growth.

It has been recently made clear that CNGC5/6/9/12 are phosphorylated and activated by OST1/SnRK2.6 for ABA-induced stomatal closure[73], and we show here that the same residues in CNGC5/6/9 are phosphorylated by CPK1 to promote RH growth. Interestingly the Arabidopsis mutant *ost1/snrk2.6* showed normal RHs. Moreover, OsCNGC9, a close homolog of Arabidopsis CNGC6 in rice, is phosphorylated and activated by OsSAPK8, a close homolog of Arabidopsis OST1, for chilling response in rice, and the site targeted by OsSAPK8 was found to be Ser645, which is localized at the C terminus rather than the N terminus[87]. Clearly, diverse protein kinases can phosphorylate and activate specific CNGCs using the same or different target sites for different biological processes. Furthermore, it seems that the

mechanisms for $Ca^{2+}$ signaling regulation vary across biological processes, cell/tissue types, and plant species.

It is well established that ROS-triggered $Ca^{2+}$ uptake plays an important role in RH growth[37]. RHD2, a NADPH oxidase, is responsible for ROS formation in RHs, and the *rhd2* mutant showed a short RH phenotype[37]. Annexin 1 was identified as a ROS-activated $Ca^{2+}$- and $K^+$-permeable channel in the plasma membrane of root epidermal cells, but the mutant *ann1* showed normal RHs[38]. It seems that Annexin 1 is not the main ROS-activated $Ca^{2+}$ channel for RH growth. In this research, we reveal the role of CPK1-CNGC modules in establishing and maintaining the $Ca^{2+}$ gradient and RH growth in Arabidopsis. However, $H_2O_2$ failed to activate CNGC6 in *Xenopus laevis* oocytes[72], suggesting that CNGCs are not the ROS-activated $Ca^{2+}$ channels in both RHs and guard cells. The ROS-activated $Ca^{2+}$ channels essential for RH growth thus remain to be identified. Nevertheless it is reasonable to speculate that both yet-to-be-identified ROS-activated $Ca^{2+}$ channels and CNGCs (ROS-insensitive $Ca^{2+}$ channels) are important for RH growth, and that

loss-of-function mutations in both types of Ca²⁺ channels will confer stronger RH phenotypes than that reported here.

RHs elongate mainly via tip growth. Interestingly it has been recently reported that shank-localized cell wall growth also contributes to RH elongation in Arabidopsis[88]. Considering that CPK1 is mainly distributed at the apex of RHs rather than in the shank area, it is very likely that the CPK1-CNGC modules are not involved in the shank-localized cell wall growth in RHs.

## Methods

### Plant materials and growth conditions

*Arabidopsis thaliana* plants (Columbia-0 ecotype) were grown in an environment-controlled growth room under a 16-h light/8-h dark daily cycle in the light of cold daylight lamps with a photon fluence rate of approximate 70−80 μmol·m⁻²·s⁻¹ during the daytime at 21 ± 1 °C, and the relative ambient humidity was ≥ 75% controlled by an ultrasonic humidifier as described[48,72]. The Arabidopsis T-DNA insertion line SALK_080155 (cpk1-1) and SALK_010530C (cpk1-2) were obtained from ABRC. The triple mutant shrh1 (short root hair1/cngc5-1 cngc6-2 cngc9-1) was generated as described previously[48]. The quadruple mutant cpk1 shrh1 was generated by crossing the single mutant cpk1-1 and triple mutant shrh1.

### Generation of transgenic Arabidopsis lines

The promoters (2 kb genomic DNA upstream of the coding regions) of CNGC5/6/9 and CPK1 were PCR amplified from Arabidopsis genomic DNA. The full length CDS (coding DNA sequences) of CPK1 and the three CNGCs were PCR amplified from Arabidopsis cDNA. The promoters and CDS were cloned into vector *pCambia1305* as described[48]. For the generation of iCNGCs (CNGC5^S20A, CNGC6^S27A, and CNGC9^S26A) and aCNGCs (CNGC5^S20D, CNGC6^S27D, and CNGC9^S26D), the point-mutations were introduced into the wild type CNGC5/6/9 by a PCR reaction. The CNGC5/6/9 point mutants were verified by sequencing. See Supplementary Data 1 for primers for the vector construction. The vectors were transformed into *Agrobacterium tumefaciens* strain *GV3101*, and the mutants cpk1-1 and shrh1 expressing the calcium indicator YC3.6[48] were transformed using floral dip method[89]. T1 seedlings were screened on ½ MS plates containing 50 mg/L hygromycin (Catalog # CH6361, Coolaber).

### RT-PCR and RT-qPCR

RT-PCR and RT-qPCR were conducted as described[72]. 7-day-old seedlings grown on ½ MS medium in a plate were ground, and total RNA was extracted from the samples using TRIzol reagent (Invitrogen, USA). RT-PCR was performed using a Hifair® V one-step RT-gDNA digestion SuperMix (Catalog # 11142ES10; Yeasen, China). RT-qPCR analysis was performed using Hieff® qPCR SYBR Green Master Mix (Catalog # 11204ES08; Yeasen, China) on a Bio-Rad CFX Connect Real-Time PCR System (Bio-Rad; USA) according to the manufacture's protocol with the primers listed in Supplementary Data 1.

### Y2-H assay

To detect the protein interactions of CPK1 with the full length CNGC5/6/9, Y2-H assays using the mbSUS system[80] were conducted. The full length CDS of *CNGC5/6/9* were cloned into the *pMetYCgate* (*Cub*) vector with *Pst* I and *Hind* III as described[73], and the CDS of *CPK1* was cloned into the *pNXgate32-3HA* (*Nub*) vector with *EcoR* I and *Sma* I. The vectors were transformed into the THY.AP4. yeast (*Saccharomyces cerevisiae*) strain using the lithium acetate transformation method[80]. The yeast clones were selected on SD/−Leu−Trp medium (Catalog # PM2220, Coolaber), and protein interactions were confirmed on selective SD/−Leu−Trp−His−Ade medium (Catalog # PM2110, Coolaber) as described[73]. Photographs were taken using a single-lens reflex (SLR) camera (EOS 7D, Canon). See Supplementary Data 1 for primer sequences.

To analyze the protein interactions of CPK1 with the fragments of CNGC5/6/9/14 (The N and C termini), Y2-H assays using the classical GAL4 Y2-H system[81] were conducted. The coding sequences of the N termini (aa1-102 for CNGC5-N, aa1-116 for CNGC6-N, aa1-117 for CNGC9-N, and aa1-101 for CNGC14-N) and the C termini (aa413-717 for CNGC5-C, aa429-747 for CNGC6-C, aa428-733 for CNGC9-C, and aa411-726 for CNGC14) of CNGC5/6/9/14 were cloned into the vector *pGBK-T7*, and the CDS of *CPK1* was cloned into the vector *pGAD-T7*, with *EcoR* I and *BamH* I, as described[73]. The vectors were co-transformed into the yeast strain *AH109* using the lithium acetate method[80]. The protein interactions of CPK1 with the transmembrane (TM) domains of CNGC5/6/9/14 were analyzed using mbSUS system. The coding sequences of the TM domains (aa103-412 for CNGC5-M, aa117-428 for CNGC6-M, aa118-427 for CNGC9-M, and aa102-410 for CNGC14-M) of CNGC5/6/9/14 were cloned into the *pMetYCgate* (*Cub*) vector with *Pst* I and *Hind* III as described[73]. Those vectors and *Nub-CPK1* were co-transformed into the THY.AP4. yeast (*Saccharomyces cerevisiae*) strain. The yeast clones of AH109 and THY.AP4 were selected on SD/-Leu-Trp medium (Catalog # PM2220, Coolaber), and the protein interactions were confirmed on selective SD/−Leu−Trp−His−Ade medium (Catalog # PM2110, Coolaber). Pictures were taken using the single-lens reflex camera (EOS 7D, Canon). Please see Supplementary Data 1 for primer sequences.

### Co-IP assay

Co-IP assays in *N. benthamiana* leaves were conducted as described[73]. The CDS of CNGC5-N, CNGC6-N, and CNGC9-N were cloned into the vector *1305-UBQ10-Venus-Flag* upstream the venus-*Flag* tag and downstream the UBQ10 promoter, and the CPK1 CDS was cloned into the vector *1305-UBQ10: Myc* upstream the *Myc* tag and downstream the *UBQ10* promoter. The vectors *CNGC5-N-Venus-Flag*, *CNGC6-N-Venus-Flag*, and *CNGC9-N-Venus-Flag* were respectively co-transformed with either *CPK1-Myc* or *mCherry-Myc* into *N. benthamiana* leaves for co-expression. Total proteins were extracted from 3 g (fresh weight) of plant samples with lysis buffer (150 mM NaCl, 1 mM EDTA, 0.05% (v/v) NP40, 1 mM PMSF, 5 mg/L Dnase, Protease Inhibitor Cocktail (Catalog # GRF101, Epizyme, China), 20 mM Tris-HCl, and pH 7.5). The extracts were incubated at 4 °C for 2 h and centrifuged at 12,000 g for 3 min at 4 °C. Anti-Myc agarose (Catalog # M20012M, Abmart) was incubated with the supernatant for 6 h at 4 °C. The Co-IP products were washed 5 times using washing buffer (150 mM NaCl, 20 mM Tris-HCl, pH 7.5). CPK1 and either mCherry or the N termini of CNGC5/6/9 were detected with anti-Myc antibodies (Catalog # Ab62928, Abcam) (1:5,000) and anti-Flag antibodies (Catalog # M20008, Abmart) (1:5,000), respectively. The chemi-luminescence signal was detected using autoradiography.

### Luciferase complementation and immuno-blot experiments in *N. benthamiana* leaves

For luciferase complementation assays in *N. benthamiana* leaves, the vectors *pCambia1300-nLUC* and *pCambia1300-cLUC* were modified. In brief, 4 x Myc was fused upstream the nLUC in the vector *pCambia1300-nLUC*, and 3 x Flag followed by a stop codon was fused downstream the cLUC in the vector *pCambia1300-cLUC*. The CDS of CPK1 and WRKY72 were cloned into the modified vector *pCambia1300-nLUC* upstream the Myc tag, while the CDS of the CNGC5, CNGC6, CNGC9, CNGC5-N, CNGC6-N, and CNGC9-N were cloned into the modified vector *pCambia1300-cLUC* downstream the cLUC and upstream the 3 x Flag. *Agrobacterium* (strain GV3101) carrying nLUC and cLUC vectors were co-injected into the *N. benthamiana* leaves. 48 hours after the injection, the *N. benthamiana* leaves were sprayed with 1 mM luciferin solution in darkness, and the luciferase luminescence was detected by taking photos with a biosystem (Model C280, Azure, USA) 7 mins after the luciferin spraying.

For immuno-blot assay in *N. benthamiana* leaves after luciferase luminescence detection, 0.2 g *N. benthamiana* leaves (fresh weight)

was ground in 300 μL extraction buffer (150 mM NaCl, 1 mM EDTA, 0.05%(v/v) NP40, 1 mM PMSF, 5 mg/L Dnase, Protease Inhibitor Cocktail (Catalog # GRF101, Epizyme, China), 20 mM Tris-HCl, and pH 7.5). The total protein was extracted from the sample in the extraction buffer as described[90]. The homogenate was centrifuged at 13,000 g for 20 min at 4 °C, and the supernatant was used for immuno-blot assay. The fused protein with nLUC was detected using anti-Myc antibody (Catalog # Ab62928, Abcam) (1:5,000), while the fused protein with cLUC was detected using anti-Flag antibody (Catalog # M20008, Abmart) (1:5,000).

### RH phenotype analysis

RH phenotypes were analyzed in Arabidopsis seedlings as described[48]. Arabidopsis seeds were surface sterilized with 0.5% sodium hypochlorite for 10 min, washed 3 times with sterile $H_2O$, and then sowed in Petri dishes containing solid medium modified from a previous method[91]. The basic solid medium contained 5 mM $KNO_3$, 1 mM $MgSO_4$, 1 mM $KH_2PO_4$, 0.1 mM NaFeEDTA, 5 mM Mes, 1% (w/v) sucrose, 0.68% (w/v) agarose, MS microelements at full strength, and pH 6.0 adjusted with Tris-HCl. 5 mM EGTA was added to the basic medium for 0 mM $Ca^{2+}$ medium. $CaCl_2$ was added to the basic medium at a concentration of 0.1 mM, 1.5 mM, and 10 mM as indicated for 0.1 mM, standard $Ca^{2+}$ (1.5 mM), and high $Ca^{2+}$ (10 mM) medium, respectively. The Petri dishes with Arabidopsis seeds were vertically placed in a growth chamber with a controlled growth condition same as that in the Arabidopsis plant growth room as described in this research (See Plant Growth in the Methods and Materials). RHs located within the 2 mm root hair zone (RHZ) of 4-5 day-old seedlings were used for RH phenotype analysis under a stereo microscope (Model M205 FCA, Leica, Germany) as described[48]. A Petri dish with 4-5 day seedlings was placed horizontally on the stage of the stereo microscope, and a set of optical sectioning images along Z axis with a 22.5 μm sectioning step and a sectioning speed of 5 steps per second were captured at room temperature (25 ± 1 °C). 20-40 optical sectioning images were captured within 4-8 sec for each seedling. The set of optical sectioning images of the bright-field were automatically and immediately merged into a 2-D photo. By that method, most RHs growing at different angles and distributed at different optical sectioning layers around the primary roots were able to be seen with consistent spatial resolution in the merged 2-D bright-field photo. The 2-D fluorescent photos merged from a set of fluorescence optical sectioning images were obtained by the same method. The length, shapes, and branching phenotypes of RHs in the merged 2-D photos were analyzed after the picture capture. No less than 10 roots with approximate 50 RHs each root were analyzed for each experimental condition each Arabidopsis line. Software GraphPad Prism 9.0 was used to plot the data.

For time-lapse analysis of RH growth, optical sectioning images were captured each 1 min for no less than 300 min at 21 ± 1 °C, and the merged 2-D photos generated immediately after optical sectioning by the software Leica Application Suite X (Leica, Germany) were used for the time-lapse analysis.

### Protein expression, purification, and in vitro phosphorylation assay

The encoding sequences of wild type CNGC9-N (aa1-117) and the mutated versions of CNGC9-N, including CNGC9-N$^{S26A}$ and CNGC9-N$^{S73A}$, were cloned into the vector *pCold-TF* downstream the TF, and this vector contained a 6×His tag upstream of TF[92]. A MBP tag was fused to the N terminus of CPK1, and the recombinant was cloned into the vector *pMAL-c5X*. The recombinants were expressed in *E. coli* strain *Rosetta-gami(DE3)pLysS* and purified using the standard protocols as described[85]. The recombinants CPK1 and CNGC9-N with the tags retained were incubated with 1 μM ATP plus 1 μCi of [γ-$^{32}$P] ATP (Catalog

# NEG502Z500UC, PerkinElmer) in a reaction buffer (20 mM Tris-HCl, 10 mM $MgCl_2$, 1 mM DTT, and pH 7.4) at 30 °C for 30 min. The reactions were terminated by boiling in SDS sample buffer and separated by 10% SDS-PAGE.

### Liquid chromatography tandem mass spectrometry (LC-MS/MS)

The target sites of CPK1 in CNGC9-N were identified by LC-MS/MS as described[93]. The recombinant GST-CPK1 and CNGC9-N were incubated with 1 mM ATP in the reaction buffer (20 mM Tris-HCl, 10 mM $MgCl_2$, 1 mM DTT, and pH 7.4) at 30 °C for 1.5 h. The reaction was diluted by adding 4 volumes of 50 mM triethylammonium bicarbonate, digested with trypsin with a 1:100 (w/w) ratio, and was incubated at 37 °C overnight. After desalting, the tryptic peptides were enriched with IMAC and analyzed by LC-MS/MS. Experiments were performed on a Q Exactive HF-X mass spectrometer coupled to Easy nLC (Thermo Fisher Scientific). The peptide mixture was loaded onto the C18-reversed phase column (25 cm long, 75 μm inner diameter) packed in-house with RP-C18 1.9 μm resin in buffer A (0.1% Formic acid in HPLC-grade water) and separated with a linear gradient of buffer B (80% acetonitrile, 0.1% Formic acid in 20% HPLC-grade water) at a flow rate of 300 nL/min. MS data was acquired using a data-dependent top-20 method by dynamically choosing the most abundant precursor ions from the survey scan (350–1800 m/z) for higher energy collision-induced dissociation (HCD) fragmentation. Determination of the target value is based on predictive Automatic Gain Control (pAGC). Dynamic exclusion duration was 45 s. Survey scans were acquired at a resolution of 60,000 at m/z 200, and resolution for HCD spectra was set to 15,000 at m/z 200. Normalized collision energy was 28 eV.

### BiFC and TEVC analysis in *Xenopus laevis* oocytes

For BiFC assays, the CDS of CPK1 was cloned into the vector *pGEMKN-YFP$^C$*, and the CDS of CNGC9 was cloned into the *pGEMKN-YFP$^N$* vector as described[78]. The cRNA were transcribed in vitro using T7 RiboMAX™ Large-scale RNA Pro-duction System (Catalog # p1712, Promega, USA). cRNA mixture was micro-injected into oocytes with 25 ng per vector in a total volume of 50 nL. The oocytes were incubated in ND96 solution at 16 ± 0.5 °C for 2-3 days after the micro-injection to allow the genes to be expressed. Images were captured using a 20× objective (NA 0.75, water) with an excitation of 487 nm and an emission of 500 to 550 nm after the incubation under a confocal microscope equipped with a detector (HyD S) and a white light laser (Stellaris 5, Leica, Germany) as described[73].

For TEVC experiments in *Xenopus laevis* oocytes, the K179R point mutation was introduced into *CPK1* by PCR, the CDS sequences of *CNGC9*, *CPK1*, and *CPK1*$^{K179R}$ were cloned into the vector *pGEMHE* with the primers listed in Supplementary Data 1, and the cRNA of *CNGC9* and the kinases were prepared in vitro using the T7 RiboMAX™ Large-scale RNA Pro-duction System (Catalog # p1712, Promega, USA). Frogs (*Xenopus laevis*) were from Shen's lab at Hangzhou Normal University (Hangzhou City, China). The oocytes were isolated from *Xenopus laevis* and microinjected with the cRNA of a single gene or cRNA mixture with an amount of 25 ng per gene in a total volume of 50 nL for each oocyte, and the oocytes microinjected with 50 nL water were used as control. The oocytes were incubated at 16 ± 0.5 °C for 2-3 days after the microinjection to allow the expression of the proteins. Channel currents were recorded using Axopatch-900A setup (Axon Instruments, USA) after the incubation, and the membrane voltage with a 2-sec duration for each voltage was stepped from -180 mV to 0 mV with a + 20 mV increment as described previously[73]. The pipette solution contained 3 M KCl. The bath solution contained 30 mM $BaCl_2$, 2 mM NaCl, 1 mM KCl, 130 mM mannitol, and 5 mM Mes-Tris (pH 5.5). Software pClampex 10.2 (Axon Instruments, USA) was used for data acquisition and analyses. Software Sigmaplot 14.0 was used to plot the data.

## Cytosolic Ca$^{2+}$ imaging and patch clamping experiments in HEK293T cells

For cytosolic Ca$^{2+}$ imaging and patch clamping experiments in HEK293T cells (Catalog # FH0242, FuHeng Biology, China), the CDS of *CPK1*, *CPK1*$^{D274A}$, wild type *CNGC5/6/9*, *iCNGCs*, and *aCNGCs* were cloned into the vector *pCI-neo-IRES-YC3.6*.

HEK293T cells were cultured in Dulbecco's Modified Eagle's Medium (DMEM) supplemented with 10% fetal bovine serum (FBS), penicillin (100 IU/mL), and streptomycin (100 μg/mL) in a water-injected incubator (Model 3111; Thermo-Fisher Scientific, USA) with 5% CO$_2$ at 37 °C in a moist atmosphere as described[82].

Vectors were transformed into HEK293T cells individually or in a combination as indicated using a Lipofectamine 2000 Transfection Reagent Kit (Invitrogen, USA), and the cells were cultured in the medium for 2-3 days to allow the gene expression. The HEK293T cells were digested with 0.25% trypsin-EDTA for no more than 1 min, collected by centrifugation at 1000 g for 1 min, washed twice by centrifugation at 1000 g for 1 min, and resuspended in a Ca$^{2+}$-free CIB solution (130 mM NaCl, 3 mM KCl, 0.6 mM MgCl$_2$, 1.2 mM NaHCO$_3$, 10 mM glucose, and 10 mM Hepes, and the pH adjusted to 7.2 with NaOH). The HEK293T cells with bright YC3.6 signal were used for Ca$^{2+}$ imaging and patch clamping experiments.

Cytosolic Ca$^{2+}$ imaging experiments in HEK293T cells were performed by monitoring the ratio of FRET/CFP of YC3.6 with a 50 ms exposure to excitation light and a 3-sec-inter-pulse period in the Ca$^{2+}$-free CIB solution under an inverted microscope (Carl Zeiss, Model D1, Germany) at room temperature (25 ± 1 °C), and 10 mM external Ca$^{2+}$ was applied as indicated. The pictures and the ratio values of interested cells/regions were automatically saved in the hard drive of a personal computer. The software MetaFluor (Version 7.8.0.0; Molecular Devises, USA) was used for data acquisition and analyses.

Patch clamping experiments in HEK293T cells were performed using the Axopatch-200B patch-clamp setup (Axon Instruments, USA) with a Digidata 1440 A digitizer (Axon Instruments, USA) combined with an inverted microscope (Model A1; Carl Zeiss, Germany) as described[82]. The bath solution contained 120 mM NMDG-Cl, 10 mM CsCl, 1 mM MgCl$_2$, 10 mM CaCl$_2$, and 10 mM Hepes, and the pH was adjusted to 7.2 with HCl. The pipette solution contained 120 mM Cs-Glutamate, 8 mM NaCl, 6.7 mM EGTA, 3.35 mM CaCl$_2$, 2 mM MgCl$_2$, 10 mM HEPES, and pH was adjusted to 7.2 with CsOH. The osmolarity was adjusted to 310 mmol kg$^{-1}$ using D-glucose for both bath and pipette solutions. Whole-cell currents were monitored and recorded by applying a 2-sec-duration ramp voltage from -180 mV to +20 mM every 12 sec for 40 times for each cell after the accession to whole cell configuration with a seal resistance of no less than 10 GΩ as described[82]. Software pClampex 10.2 (Axon Instruments, USA) was used for data acquisition and analysis. Software Sigmaplot 14.0 was used to plot the data.

## Cytosolic Ca$^{2+}$ imaging analysis in RHs

Triple mutant *shrh1* expressing YC3.6 was generated as described[48]. YC3.6 was introduced into the mutant *cpk1-1* by crossing the mutant to the wild type plants expressing *YC3.6*, and the homozygous *cpk1-1* mutant with bright YC3.6 signal in RHs were isolated from the progeny plants. The Arabidopsis transgenic lines were generated based on the wild type and mutants expressing *YC3.6* in RHs. FRET/CFP-based cytosolic Ca$^{2+}$ imaging analysis were performed in RHs of 4-day-old seedlings adhered to a cover glass with the medical adhesive glue (Hollister, USA) under the inverted microscope (Model Dl; Carl Zeiss, Germany). The cover glass with roots and RHs was immersed in the working solution (5 mM KCl, 100 μM CaCl$_2$, 10 mM Mes-Tris, pH 5.7) in a chamber. The ratio of FRET/CFP of YC3.6 was monitored with a 50 ms exposure to excitation light and a 3-sec-inter-pulse period, and the data were automatically saved in the hard drive of a personal computer. In situ calibration was performed as described[94].

## Statistical analysis

Student's *t*-test and one-way ANOVA were used to determine significant differences between samples. No data were excluded from all analyses. The investigators were not blinded to allocation during experiments and outcome assessment.

## Reporting summary

Further information on research design is available in the Nature Portfolio Reporting Summary linked to this article.

# Data availability

All data are incorporated into the article and its online supplementary materials. The LC-MS/MS data generated in this study have also been deposited in the ProteomeXchange Consortium via the PRIDE partner repository with an accession code PXD055717. Gene sequence data in this article can be found in GenBank database using the accession numbers AT5G04870 (*CPK1*), AT5G57940 (*CNGC5*), AT2G23980 (*CNGC6*), AT4G30560 (*CNGC9*), and AT2G24610 (*CNGC14*). Source data are provided with this paper.

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

## Acknowledgements

We thank Dr. Shanshan Wang and Dr. Wenjuan Cai (CAS Center for Excellence in Molecular Plant Sciences, CAS, Shanghai 200032, China) for assistance in LC-MS/MS and BiFC experiments, respectively, and Dr. Jeremy Murray (CAS Center for Excellence in Molecular Plant Sciences, CAS, Shanghai 200032, China) for a critical reading of this article.
**Funding** This research was mainly supported by the National Natural Science Foundation of China (grant numbers 32470332 and 32270279 to W.Y.-F., 32300253 to T.Y.-Q., and 32400214 to Y.Y.) and the Key Laboratory of Plant Carbon Capture of CAS (CNXT002 to W.Y.-F.), and was partially supported by the Postdoctoral Fellowship Program of CPSF (GZB20240744 to Y.Y.), the China Postdoctoral Science Foundation (2024M753231 to Y.Y.), and Shanghai Post-doctoral Excellence Program (2024652 to Y.Y.).

## Author contributions

Z.M. screened the Arabidopsis mutant library, constructed most vectors, generated quadruple mutants *cpk1 shrh1* and transgenic Arabidopsis lines, and performed most experiments, including Y2-H, luciferase complementation, Co-IP, and cytosolic $Ca^{2+}$ imaging experiments in both HEK293T cells and RHs. D.B.-Y. performed RH time-lapse, BiFC, TEVC, and patch clamping experiments. T.Y-Q. and Z.M. performed

LC-MS/MS experiments and identified the CPK1-target sites. Y.Y. and Z.Y. were involved in the vector construction. W.Y.-F. proposed the project, designed the experiments and wrote the manuscript.

## Competing interests

The authors declare no competing interests.
