## [Peer Review File · Nature Communications]

REVIEWER COMMENTS

Reviewer #1 (Remarks to the Author):

In order to identify kinases that are essential for root hair development, the authors screened a mutant library and identified *cpk1*, which displayed a similar phenotype as the triple CNGC mutant *shrh-1*. Through a number of experiments, they show that CPK1 acts upstream of these CNGCs in RH development. They also confirmed the physical interaction of CPK1 and the N-termini of CNGC5/6/9 by several methods. They also showed that CPK1 activates CNGCs in *Xenopus* oocytes and HEK cells. Finally, they identified phosphorylation sites in CNGC5/6/9 and confirmed that the phospho-dead versions could not be activated by CPK1. The *cpk1* mutant could be rescued by the expression of a phospho-mimic but not a phospho-dead version of either CNGC. Overexpression of phospho-dead versions of CNGCs also lead to a *shrh1* phenotype. The manuscript provides a comprehensive analysis connecting CPK1 to CNGC5/6/9 in root hair development.

I have some questions and comments:

-Expression of each one of the phospho-mimic versions of the CNGCs alone can overcome the *cpk1* phenotype (Fig 8). What is the author's explanation for that? On the other hand, the single *cngc* mutants do not show a phenotype. Are the 3 channels forming a heteromeric channel and activation of either one activates the channel? What is the model for these 3 CNGCs? Have the authors tested the subunit interaction?

-Binding assays: It would be good to include another CNGC as a negative control.

-Overexpressing CPK1 in *shrh1* does not completely suppress the RH branching phenotype (FIG 2C, neither does CNGC9:OX in *cpk1*). Do the authors have an explanation for that?

- Does CPK1 also phosphorylate CNGC14, which has also been implicated in RH development? It would also be good to discuss about CNGC14' role in the discussion.

Minor issues:

L 37 the nonphosphomimicking CNGG5/6/9 with an S-to-A point...

- Sounds strange and confusing, usually we say phospho-dead.

There are a few sloppy sentences in the introduction. Please double check, e.g.

L 113 an intrinsic regulatory machinery, which is composed of diverse components, including Ca²⁺, ROS, pH, cytoskeleton, ROP et al. External Ca²⁺ influx occurs during RH initiation

-maybe factors is better than component here.

L 121 During RH growth, the cytosolic Ca²⁺ oscillation is highly coordinated with and interacts to other regulating molecules, including ROS, pH, ROPs, and

-pH is not a molecule

Reviewer #2 (Remarks to the Author):

In this manuscript, authors uncovered CPK-CNGC module is important for Ca²⁺ transport and root hair growth. Overall, I think that the experiments and conclusions are valid. However, at the current state of writing, this paper is unacceptable for publication and requires extensive revision by a native English speaker. Specific points are below:

1. Line 36: “phosphomimicking” should be phosphomimic.

2. Line 37: instead of “nonphosphomimicking” I suggest using “phospho-dead”

3. Line 39: “required and sufficient” is usually written as “necessity and sufficient” in English. Line 185-186 as well. Line 522 as well.

4. Line 45: “The” root system.

Line 46-47 has a run-on sentence: “...microbes, and root hairs (RHs), the tubular-shaped...” is better as two sentences: “...microbes. Root hairs (RHs), the tubular-shaped...”

Line 50: to what is “plant climbing” referring? Does the author mean “support of the aerial parts of the plant” or something similar. In brief, the manuscript needs to be polished by a native English speaker. There are numerous more examples in the manuscript to which I will try not to itemize.

Line 71- 73. Just terrible writing. “The development of RHs can be roughly categorized into three steps: cell-fate determination of the root epidermal cells, and the initiation and elongating tip growth of RHs.” Is written as two steps! Try “The development of RHs can be roughly categorized into three steps: cell-fate determination of the root epidermal cells, the initiation of RHs, and subsequent elongation through tip growth.

Line 165: In formal English writing (like scientific publications) Sentences do not begin with conjunctions (and/but/or)!

Line 182-187 is one long run-on sentence.

Line 260 remove “on the”

Line 308-312: the word “condition” is used 8 times in one sentence!

5. Line 108: “This process is referred to as tip growth.” While tip growth is the primary method of RH elongation, there exists another method which should be cited for comprehensiveness: Herburger, K., Schoenaers, S., Vissenberg, K. et al. Shank-localized cell wall growth contributes to Arabidopsis root hair elongation. *Nat. Plants* 8, 1222–1232 (2022). <https://doi.org/10.1038/s41477-022-01259-y>. However, based on the tip-localized calcium gradient, this mechanism probably does not come into play with CPK1, though it might....

6. Line 191-192: “RH growth” is better than RH development in line 191-192 and the whole manuscript.

7. Line 192-196: All the T-DNA kinase mutants analyzed in this screen should be put into supplemental information as a resource for the community. Also in discussion of Line 615-617, author should have put a results in supplementary information.

8. Lines 197-202 should go methods section, as well as lines 251-254. The authors need to carefully check what belongs in the results part, and move the detailed methods description into the methods section.

9. Line 212-214: RT-qPCR to characterize the KO mutant should be in the results at the beginning of the phenotyping section.

10. Line 283, “conforred” is wrongly spelled word.

11. In the results, while a summary of each section is appreciated. A single sentence does not constitute a paragraph.

12. Line 416 -417: suggesting that the Ser26 was the main targeting site of CPK1 on CNGC9-N; compared to line 427: “corresponding CPK1-targeted Ser27 in CNGC9”

13. Line 419-420: “was strongly inhibited in the absence of Ca²⁺ (without Ca²⁺ added, but with 5 mM EGTA added) (Fig. 6c).” In Fig. 6c and Methods section no mention is made of the use of EGTA.

14. Line 480-481 is a direct repeat of Line 436-437 and should be deleted.

15. Section: “The RH phenotypes of the triple mutant *shrh1* and single mutant *cpk1-1* are fully rescued by the expression of aCNGCs rather than iCNGCs under the respective CNGC native promoters” Really needs to be re-written, especially since in the heading the reader has no idea to which “aCNGCs” and “iCNGCs” refer. These should at least be defined in the first sentences of this section more clearly. As an example: “To simplify nomenclature, we now designate CNGC5S20A, CNGC6S27A, and CNGC9S26A488, as iCNGCs (inactivated CNGCs), and CNGC5S20D, CNGC6S27D, and CNGC9S26D as aCNGCs (activated CNGCs) [ref].

16. Line 505: “...to be rescued in the transgenic lines CNGCS20A(*shrh1*)” should be CNGC5S20A

17. Line 517, higher RH, not higher HR.

18. Line 625: I do not like the word “undetectable.” I suggest author use “condition-specific”

19. In the Y-axis of Figure 1f, it should be elongation, not “elangement.”

20. In figure 3 and 4 the Split-luc assays need western blots as controls, need more controls (as mentioned in Zhou et al., 2018, the authors who made the split-luc vector).

21. In figure 4c, why author did not perform co-ip assays for CPK1 with CNGC5 and CNGC 6.

22. Can author compare the phosphorylation abundance of CNGC5/6/9 at S26 and S73 sites in cpk1 mutant and WT? The current manuscript only has in vitro information, but not in vivo. Fig 6C shows massive phosphorylation with S73A (higher than CNGC9-N). Are there physiological implications for this, as there is still residual phosphorylation on S26A?

24. In Supplementary Fig. 5, under a high concentration of Ca²⁺, the mutants did not show as clear a phenotype as under the 1.5mM Ca²⁺ condition. The authors did not offer a clear explanation in the discussion.

25. In the Discussion section of the manuscript, the authors write too much about results themselves rather than discussing the results in a broader sense and their implications for RH elongation. Again, the English is so poor in places that I am unable to draw a meaningful conclusion to what point is actually meant.

Reviewer #3 (Remarks to the Author):

The present work builds upon a previous study by the authors' group (Tan, Plant Communications, 2019) that demonstrated the necessity of CNGC5, CNGC6, and CNGC9 in root hair growth. In the current manuscript, the authors provide evidence that CPK1 phosphorylates and activates CNGC5/6/9 to regulate root hair development, identifying the main phosphorylation sites involved. The genetic evidence presented is robust and compelling, and the overall findings are intriguing. However, I have several concerns that should be addressed.

1. In Figure 9, the cytosolic calcium levels in the cpk1 and shrh1 mutants are comparable, which raises a significant question. Previous studies from the authors' group have established that CNGC5/6/9 function as active calcium channels (as shown in Figure 5 of this work and in Tan, Plant Communications, 2019). In the shrh1 mutant, CNGC5/6/9 are knocked out, leading to diminished cytosolic Ca²⁺ gradients, which is reasonable. However, in the cpk1 mutant, CNGC5/6/9 remains intact and should exhibit basal activity, contributing to cytosolic calcium signaling to some extent. This discrepancy suggests an important issue that must be considered: Plant CNGCs possess calmodulin-binding domains (CaMBD) and IQ domains that can interact with calmodulin (CaM). Previous studies have indicated that CaM can inhibit the activity of CNGC channels. Therefore, it is essential to first test whether CaM inhibits the activity of CNGC5/6/9. If so, the authors should then investigate whether CPK1 activates the CNGC5/6/9-CaM channel complex. This aspect should not be overlooked.

2. In line 192, the information regarding the kinase mutant library is unclear. It would be beneficial to provide a detailed composition of this mutant library to enhance the clarity of the methodology.

3. The manuscript lacks a clear explanation of how CNGC5/6/9-mediated calcium influx results in the formation of a gradient of cytosolic calcium. Do CNGC5/6/9 exhibit polarized localization in the root hair tip? This information is critical for understanding the functional implications of the calcium gradient.

4. The methodology for two-electrode voltage clamp (TEVC) recordings conducted on *Xenopus* oocytes is not described in the manuscript.

I look forward to seeing these revisions implemented.

REVIEWER COMMENTS AND OUR RESPONSES

Reviewer #1 (Remarks to the Author):

In order to identify kinases that are essential for root hair development, the authors screened a mutant library and identified *cpk1*, which displayed a similar phenotype as the triple CNGC mutant *shrh-1*. Through a number of experiments, they show that CPK1 acts upstream of these CNGCs in RH development. They also confirmed the physical interaction of CPK1 and the N-termini of CNGC5/6/9 by several methods. They also showed that CPK1 activates CNGCs in *Xenopus* oocytes and HEK cells. Finally, they identified phosphorylation sites in CNGC5/6/9 and confirmed that the phospho-dead versions could not be activated by CPK1. The *cpk1* mutant could be rescued by the expression of a phospho-mimic but not a phospho-dead version of either CNGC. Overexpression of phospho-dead versions of CNGCs also lead to a *shrh1* phenotype. The manuscript provides a comprehensive analysis connecting CPK1 to CNGC5/6/9 in root hair development.

I have some questions and comments:

-Expression of each one of the phospho-mimic versions of the CNGCs alone can overcome the *cpk1* phenotype (Fig 8). What is the author's explanation for that? On the other hand, the single *cngc* mutants do not show a phenotype. Are the 3 channels forming a heteromeric channel and activation of either one activates the channel? What is the model for these 3 CNGCs? Have the authors tested the subunit interaction?

Response: We thank the reviewer for raising this concern.

It has been reported that CNGC family members are capable of forming homo- and hetero-tetramers, which function as Ca^{2+} channels in plant cells (Chiasson et al., eLife 6: e25012, 2017). The protein-protein interactions of CNGC6 to multiple CNGC members, including CNGC6 itself, have been analyzed and observed (Tan YQ and Yang Y et al., Plant Cell 35: 239-259, 2023). It has been established that the function of tetramer channels from Shaker family can be dominant-negatively inhibited by the integration of loss-of-function subunits (Kwak et al., Plant Physiol 127: 473-485). Similarly it has been recently reported that the multiple CNGCs can be dominant-negatively inhibited by the over expression of loss-of-function CNGC6 in guard cells (Tan YQ and Yang Y et al., Plant Cell 35: 239-259, 2023). These previous studies together demonstrated that CNGCs are capable of forming homo- and hetero-tetramers. So the reviewer is correct that the CNGCs can form tetramers with a strong redundancy. We added a sentence into the MS to introduce the background, so that the readers can have a better understanding about the results in this research (please see lines 156-157).

It has been well-established that multiple CNGCs function as Ca^{2+} channels in both RHs (Tan YQ et al., Plant Commun 1: 100001, 2020) and guard cells (Tan YQ and Yang Y et al., Plant Cell 35: 239-259, 2023) with a strong redundancy. The single mutants *cngc5-1*, *cngc6-2*, and *cngc9-1* showed no RH phenotype, the triple mutant *shrh1* showed strong RH phenotypes, and the complementary expression of a single wild type CNGC is sufficient to fully rescue the RH phenotypes of *shrh1* (Tan YQ et al., Plant Commun 1: 100001, 2020). These reports have established the strong redundancy between multiple CNGCs in RHs.

It has been reported that the phospho-mimic versions of CNGCs, including CNGC5^{S20D}, CNGC6^{S27D}, CNGC9^{S26D}, showed a large Ca^{2+} channel activity, similar to the

OST1-mediated phosphorylation-induced activation of wild type CNGC5/6/9 (Yang Y et al., Plant Cell 36: 2328-2358, 2024). We noticed that the OST1-target sites are identical to the main CPK1-target sites for CNGC5/6/9. Then CNGC5^{S20D}, CNGC6^{S27D}, CNGC9^{S26D} are three constitutively activated Ca²⁺ channels, and have been designated as aCNGCs (activated CNGCs) (Yang Y et al., Plant Cell 36: 2328-2358, 2024).

Taken all these previous reports and the results in this research together, the RH phenotypes of both *cpk1* and *shrhl* mutants are caused by the lack of activated CNGCs in RHs. Thus the expression of a single phospho-mimic CNGC, a constitutive activated CNGC version, is sufficient to fully rescue the RH phenotypes of both *cpk1* and *shrhl*. We added a sentence to the MS to explain this, and cited the related references (please see lines 481-483, and 499-500).

-Binding assays: It would be good to include another CNGC as a negative control.

Response: We agree with the reviewer. It has been reported that CNGC14 is highly expressed in Arabidopsis root hairs, but the triple mutant *cngc5-1 cngc6-2 cngc9-1* (*shrhl*, *short root hair 1*) and the quadruple mutant *cngc5-1 cngc6-2 cngc9-1 cngc14-1* (*shrh2*, *short root hair 2*) show quite similar root hair phenotypes (Tan YQ et al., Plant Commun 1: 100001, 2020), suggesting that CNGC14 is not important for root hair growth. We think CNGC14 is an ideal negative control. We then conducted further Y2-H assay to test the protein interactions of CPK1 to the fragments of CNGC14 (including its N and C termini and transmembrane domain), and did not observe any obvious protein interaction. These new data were added to Fig. 4a. We described the new data in the MS accordingly (please see lines 339-344).

It has been reported that WRKY72 is involved in RH growth. We then used WRKY72 as another negative control. We conducted further luciferase complementation assay in *N. benthamiana* leaves to test the protein interactions of WRKY72 to the full-length CNGC5/6/9 and their fragments, and did not observe any obvious protein interaction. These new data were added to Fig. 3b and Fig. 4b. We described the new results in the MS accordingly (please see lines 319-321 and 346-352).

-Overexpressing CPK1 in *shrh1* does not completely suppress the RH branching phenotype (FIG 2C, neither does CNGC9:OX in *cpk1*). Do the authors have an explanation for that?

Response: The overexpression of *CPK1* in the *shrh1* background did not rescue RH phenotypes, and also did not confer longer RH phenotype, compared to wild type plants. According to our hypothesis that CPK1-mediated phosphorylation of CNGC5/6/9 is necessary and sufficient for RH growth. Thus, this failure should be caused by the missing of CNGC5/6/9, the downstream targets of CPK1, in the transgenic line *CPK1-OE3(shrh1)*. The overexpression of *CNGC9* in the background of *cpk1-1* failed to rescue the RH phenotypes of *cpk1* mutant, and also failed to confer longer RHs than that of wild type. Similarly, this should be caused by the failure of CPK1-mediated phosphorylation of CNGCs because of the lack of CPK1 in the *cpk1* mutant background. We added an explanation in the MS so that the readers can understand the data better (please see lines 289-293).

- Does CPK1 also phosphorylate CNGC14, which has also been implicated in RH development? It would also be good to discuss about CNGC14' role in the discussion.

Response: We tested the protein-protein interaction between CPK1 and CNGC14, and the data showed that CPK1 does not interact with the N and C termini and the transmembrane domain of CNGC14. These new data were added to Fig. 4a as negative controls. We described the new data in the MS (see lines 339-344). Moreover, the triple mutant *shrh1 (cngc5/6/9)* and the quadruple mutant *shrh2 (cngc5/6/9/14)* showed similar root hair phenotype (Tan YQ et al., Plant Commun 1: 100001, 2020). These data in this research and the previous report together suggest that CNGC14 is not the downstream target of CPK1 in Arabidopsis RHs. We discussed the role of CNGC14 in RHs as the reviewer suggested (please lines 570-577).

Minor issues: L 37 the nonphosphomimicking CNGC5/6/9 with an S-to-A point...
- Sounds strange and confusing, usually we say phospho-dead.

Response: We thank the reviewer for the suggestion. We have changed the word nonphosphomimicking to phospho-dead across the whole MS.

There are a few sloppy sentences in the introduction. Please double check, e.g. L 113 an intrinsic regulatory machinery, which is composed of diverse components, including Ca²⁺, ROS, pH, cytoskeleton, ROP et al. External Ca²⁺ influx occurs during RH initiation -maybe factors is better than component here.

Response: We thank the reviewer for the nice suggestion. We have changed the word “components” to “factors” (please see line 107).

L 121 During RH growth, the cytosolic Ca²⁺ oscillation is highly coordinated with and interacts to other regulating molecules, including ROS, pH, ROPs, and -pH is not a molecule.

Response: We thank the reviewer for pointing this error out. We changed the word “molecules” to “factors” (please see line 117).

Reviewer #2 (Remarks to the Author):

In this manuscript, authors uncovered CPK-CNGC module is important for Ca²⁺ transport and root hair growth. Overall, I think that the experiments and conclusions are valid. However, at the current state of writing, this paper requires extensive revision by a native English speaker. Specific points are below:

Response: We thank the reviewer for the comments on our MS and the suggestions in writing. We carefully edited the MS, and corrected the mistakes as the reviewer suggested. We also sent our MS to a colleague who is a native English speaker for a critical reading. The whole MS was polished by the colleague, and the writing is much better.

1. Line 36: “phosphomimicking” should be phosphomimic.

Response: “Phosphomimicking” was changed to “phospho-mimic” across the whole MS. We thank the reviewer for the suggestion.

2. Line 37: instead of “nonphosphomimicking” I suggest using “phospho-dead”

Response: “nonphosphomimicking” was changed to “phospho-dead”. We appreciate for the suggestion.

3. Line 39: “required and sufficient” is usually written as “necessity and sufficient” in English. Line 185-186 as well. Line 522 as well.

Response: We thank the reviewer for the suggestion. The related sentences were revised by the native English speaker during his MS polishing.

4. Line 45: “The” root system. Line 46-47 has a run-on sentence: “...microbes, and root hairs (RHs), the tubular-shaped...” is better as two sentences: “...microbes. Root hairs (RHs), the tubular-shaped...”

Response: We split the long sentence into two short sentences as the reviewer suggested. Please see lines 42-45.

Line 50: to what is “plant climbing” referring? Does the author mean “support of the aerial parts of the plant” or something similar. In brief, the manuscript needs to be polished by a native English speaker. There are numerous more examples in the manuscript to which I will try not to itemize.

Response: It has been reported that the roots of English ivy (*Hedera helix*) plants exude a yellowish mucilage that promotes the capacity of these plants to climb vertical surfaces, and RHs play an important role in the plant climbing (Huang Y et al., PNAS 2016 113: E3193-202). We apologize for the confusing description. We revised the description accordingly to improve the description to avoid any confusion (please see lines 46-48).

The MS was polished by a Canadian colleague, a native English speaker.

Line 71- 73. Just terrible writing. “The development of RHs can be roughly categorized into three steps: cell-fate determination of the root epidermal cells, and the initiation and elongating tip growth of RHs.” Is written as two steps! Try “The development of RHs can be roughly categorized into three steps: cell-fate determination of the root epidermal cells, the initiation of RHs, and subsequent elongation through tip growth.”

Response: We revised the sentence as the reviewer suggested (please see lines 68-70).

Line 165: In formal English writing (like scientific publications) Sentences do not begin with conjunctions (and/but/or)!

Response: We merged two short sentences into a little bit longer sentence to avoid the grammar mistake. Similar grammar mistakes were also corrected across the whole MS, and were double checked by a native English speaker.

Line 182-187 is one long run-on sentence.

Response: We split the long sentence into a few shorter sentences, and also revised the description accordingly. Please see lines 174-180.

Line 260 remove “on the”

Response: The two words were removed (please see line 253).

Line 308-312: the word “condition” is used 8 times in one sentence!

Response: We removed the description about the medium from the Results and modified the description about the medium in Methods (see lines 722-728).

5. Line 108: “This process is referred to as tip growth.” While tip growth is the primary method of RH elongation, there exists another method which should be cited for comprehensiveness: Herburger, K., Schoenaers, S., Vissenberg, K. et al. Shank-localized cell wall growth contributes to Arabidopsis root hair elongation. *Nat. Plants* 8, 1222–1232 (2022). <https://doi.org/10.1038/s41477-022-01259-y>. However, based on the tip-localized calcium gradient, this mechanism probably does not come into play with CPK1, though it might....

Response: We thank the reviewer for pointing this out. We have noticed the nice work by Herburger et al. (Nat Plants 2022). We agree with the reviewer that CPK1 may not be involved in the shank-localized cell wall growth. We discussed the possibility, and cited the reference in the MS as the reviewer suggested in the Discussion (please see lines 618-622).

6. Line 191-192: “RH growth” is better than RH development in line 191-192 and the whole manuscript.

Response: We agree with the reviewer, and have changed “RH development” to “RH growth” across the whole MS.

7. Line 192-196: All the T-DNA kinase mutants analyzed in this screen should be put into supplemental information as a resource for the community. Also in discussion of Line 615-617, author should have put a results in supplementary information.

Response: A list of the Arabidopsis mutants screened in this research was added to the MS as Supplementary Data Set 1, and we cited the supplementary material in lines 200 and 597 as the reviewer suggested.

8. Lines 197-202 should go methods section, as well as lines 251-254. The authors need to carefully check what belongs in the results part, and move the detailed methods description into the methods section.

Response: We agree with the reviewer that detailed description about materials and methods should go to Methods and Materials. We feel the optical sectioning technique using a stereo microscope is unique, and is also the key to successfully identify the mutants *cpk1-1* and *cpk1-2* with strong RH phenotypes. Thus, a brief introduction about this technique and performance is helpful and convenient for readers to understand the results without reading the Methods first. For this reason, we prefer to introduce the performance very briefly for only once before we start to describe the plant phenotypes. Please see lines 190-193.

We double checked the MS carefully, and removed the unnecessary method-description from Results.

9. Line 212-214: RT-qPCR to characterize the KO mutant should be in the results at the beginning of the phenotyping section.

Response: We adjusted the sequence of Supplementary Figures, and the RT-qPCR data are used as Supplementary Fig. 3 now.

10. Line 283, “conforred” is wrongly spelled word.

Response: This typing error has been corrected (line 273). Thank you.

11. In the results, while a summary of each section is appreciated. A single sentence does not constitute a paragraph.

Response: These errors were corrected across the whole MS. Thank you.

12. Line 416 -417: suggesting that the Ser26 was the main targeting site of CPK1 on CNGC9-N; compared to line 427: “corresponding CPK1-targeted Ser27 in CNGC9”

Response: Our colleague, the native English speaker, who polished this MS, revised the description as “CPK1-target site” across the whole MS. Both the reviewer and our colleague are native English speakers, we thus believe that both “CPK-target site” and “CPK1-targeted site” are correct for the description. We retained the English editing by our colleague, and wish this is OK for the reviewer.

13. Line 419-420: “was strongly inhibited in the absence of Ca²⁺ (without Ca²⁺ added, but with 5 mM EGTA added) (Fig. 6c).” In Fig. 6c and Methods section no mention is made of the use of EGTA.

Response: We thank the reviewer for pointing out this error. The description about the 0 Ca²⁺ (no Ca²⁺ added, with 5 mM EGTA added) medium was added to the MS. The description about all the different medium was corrected accordingly (please see lines 722-728).

14. Line 480-481 is a direct repeat of Line 436-437 and should be deleted.

Response: The repeating sentence was deleted.

15. Section: “The RH phenotypes of the triple mutant shrh1 and single mutant cpk1-1 are fully rescued by the expression of aCNGCs rather than iCNGCs under the respective CNGC native promoters” Really needs to be re-written, especially since in the heading the reader has no idea

to which “aCNGCs” and “iCNGCs” refer. These should at least be defined in the first sentences of this section more clearly. As an example: “To simplify nomenclature, we now designate CNGC5S20A, CNGC6S27A, and CNGC9S26A488, as iCNGCs (inactivated CNGCs), and CNGC5S20D, CNGC6S27D, and CNGC9S26D as aCNGCs (activated CNGCs) [ref].

Response: We agree with the reviewer. We added a paragraph to define the iCNGCs and aCNGCs, and cited a reference as the reviewer suggested (please see lines 472-480).

16. Line 505: “...to be rescued in the transgenic lines CNGCS20A(shrh1)” should be CNGC5S20A

Response: The mistake was corrected, please see line 495. Thank you.

17. Line 517, higher RH, not higher HR.

Response: The related sentence has been revised, and the mistake was avoided. Thank you.

18. Line 625: I do not like the word “undetectable.” I suggest author use “condition-specific”

Response: The Discussion has been rewritten, and the word “undetectable” and the related sentence have been removed from the MS.

19. In the Y-axis of Figure 1f, it should be elongation, not “elangement.”

Response: The error has been corrected. Thank you.

20. In figure 3 and 4 the Split-luc assays need western blots as controls, need more controls (as mentioned in Zhou et al., 2018, the authors who made the split-luc vector).

Response: WRKY72 was used as negative controls in luciferase complementation assays. No protein interaction of WRKY72 to the full length CNGC5/6/9 and their N termini was observed. The new results were added to the MS as Fig. 3b and Fig. 4b. Accordingly, we conducted western blot assays to verify the expression of the proteins in *N. benthamiana* leaves to support our luciferase complementation assays. The immuno-blot results were added to the MS as Fig. 3c and Fig. 4c. We described the data in the MS accordingly (please see lines 319-323 and 346-352).

21. In figure 4c, why author did not perform co-ip assays for CPK1 with CNGC5 and CNGC 6.

Response: We conducted further Co-IP assay, and the protein interactions of CPK1 with both CNGC5-N and CNGC6-N were observed. The new results were added to the MS as Fig. 4d and Fig. 4e. We described the new data in the MS accordingly (please see lines 353-355).

22. Can author compare the phosphorylation abundance of CNGC5/6/9 at S26 and S73 sites in cpk1 mutant and WT? The current manuscript only has in vitro information, but not in vivo. Fig 6C shows massive phosphorylation with S73A (higher than CNGC9-N). Are there physiological implications for this, as there is still residual phosphorylation on S26A?

Response: Our data showed that Ser 20 in CNGC5, Ser27 in CNGC6, and Ser26 in CNGC9 are the main CPK1-target sites. We cannot exclude the possibility that the Ser 58, Ser73, and Ser73 play minor roles as the secondary CPK1-target sites. We thus designated S20 in CNGC5, S27 in CNGC6 and S26 in CNGC9 as the three main CPK1-target sites, and designated S58 in CNGC5 and S73 in both CNGC6 and CNGC9 as the secondary CPK1-target sites. To test the possible function of the secondary CPK1-target sites *in vivo*, we generated point-mutated CNGC5/6/9 by substituting the secondary CPK1-target sites with either A or D, and the mutated versions were designated as CNGC5^{S58A}, CNGC5^{S58D}, CNGC6^{S73A}, CNGC6^{S73D}, CNGC9^{S73A}, and CNGC9^{S73D}. We expressed the mutated CNGCs under the respective CNGC native promoters in the background of *cpk1-1* mutant, and the transgenic lines were selected and designated as CNGC5^{S58A}(*cpk1-1*) (expressing CNGC5^{S58A} under CNGC5 native promoter in the background of *cpk1-1*), CNGC5^{S58D}(*cpk1-1*), CNGC6^{S73A}(*cpk1-1*), CNGC6^{S73D}(*cpk1-1*), CNGC9^{S73A}(*cpk1-1*), and CNGC9^{S73D}(*cpk1-1*). We analyzed the RH phenotypes, and only observed *cpk1-1*-like RH phenotypes in those transgenic lines, including shorter RHs and more RH branching, compared to wild type. These new results suggested that the secondary CPK1-target sites are not essential for RH growth in Arabidopsis. The new results were added to the MS as Supplementary Fig. S10). We described the new results in the MS accordingly (please see lines 526-541).

24. In Supplementary Fig. 5, under a high concentration of Ca²⁺, the mutants did not show as clear a phenotype as under the 1.5mM Ca²⁺ condition. The authors did not offer a clear explanation in the discussion.

Response: We added an explanation of the phenotypes under both low and high Ca²⁺ conditions in Discussion as the reviewer suggested (please see lines 578-586).

25. In the Discussion section of the manuscript, the authors write too much about results themselves rather than discussing the results in a broader sense and their implications for RH elongation. Again, the English is so poor in places that I am unable to draw a meaningful conclusion to what point is actually meant.

Response: We rewrote the whole Discussion to avoid un-necessary description of our own data.

Reviewer #3 (Remarks to the Author):

The present work builds upon a previous study by the authors' group (Tan, Plant Communications, 2019) that demonstrated the necessity of CNGC5, CNGC6, and CNGC9 in root hair growth. In the current manuscript, the authors provide evidence that CPK1 phosphorylates and activates CNGC5/6/9 to regulate root hair development, identifying the main phosphorylation sites involved. The genetic evidence presented is robust and compelling, and the overall findings are intriguing. However, I have several concerns that should be addressed.

1. In Figure 9, the cytosolic calcium levels in the *cpk1* and *shrh1* mutants are comparable, which raises a significant question. Previous studies from the authors' group have established that CNGC5/6/9 function as active calcium channels (as shown in Figure 5 of this work and in Tan, Plant Communications, 2019). In the *shrh1* mutant, CNGC5/6/9 are knocked out, leading to diminished cytosolic Ca²⁺ gradients, which is reasonable. However, in the *cpk1* mutant, CNGC5/6/9 remains intact and should exhibit basal activity, contributing to cytosolic calcium signaling to some extent. This discrepancy suggests an important issue that must be considered: Plant CNGCs possess calmodulin-binding domains (CaMBD) and IQ domains that can interact with calmodulin (CaM). Previous studies have indicated that CaM can inhibit the activity of

CNGC channels. Therefore, it is essential to first test whether CaM inhibits the activity of CNGC5/6/9. If so, the authors should then investigate whether CPK1 activates the CNGC5/6/9-CaM channel complex. This aspect should not be overlooked.

Response: We thank the reviewer for raising this concern. We agree with the reviewer that CNGCs may have some basal activity in RHs in the absence of CPK1. In fact we observed a small activity of iCNGCs with an S-to-A point-mutation at the main CPK1-target sites compared to mock control (Fig. 7). However the expression of iCNGCs under their respective native promoters in the background of *cpk1* and *shrh1* cannot obviously restore the cytosolic Ca²⁺ gradient at RH apex (Fig. 9) and rescue the RH phenotypes (Fig. 8), suggesting that the basal activity of iCNGCs is not sufficient to establish and maintain the sharp Ca²⁺ gradient and RH growth.

A previous study reported that CNGC14 plays an important role in touch sensing in RH (Zhang et al., Mol Plant 10: 1004-1006, 2017), and the overexpression of *CaM7* can inhibit CNGC14 and confer a short RH phenotype (Zeb et al., J Integr Plant Biol 62: 887-896, 2020). However, the functions of calmodulin in RH growth are not supported by genetic evidence. We feel that the short RH phenotype of the CaM7-overexpression line is an effect of the artificial over-accumulation of CaM7 protein in RHs. To test whether calmodulins function in RHs, we collected a number of *cam* mutants, including single mutants *cam1*, *cam2*, *cam3*, *cam4*, *cam5*, *cam6*, and *cam7*, and double mutant *cam1 cam4* (*cam1/4*). We analyzed the RH phenotypes, and did not observe any obvious RH phenotype across all the *cam* mutants, suggesting that CaMs are not essential for RH growth in Arabidopsis. We added the new data into the MS as Supplementary Fig. 1, and described the research background and our new data in the MS accordingly (please see lines 184-196).

2. In line 192, the information regarding the kinase mutant library is unclear. It would be beneficial to provide a detailed composition of this mutant library to enhance the clarity of the methodology.

Response: We added a mutant list to the MS as Supplementary Data Set 1, and calmodulin mutants and all the kinase mutants screened in this study were included. We also cited Supplementary Data Set 1 in the MS (lines 200 and 597).

3. The manuscript lacks a clear explanation of how CNGC5/6/9-mediated calcium influx results in the formation of a gradient of cytosolic calcium. Do CNGC5/6/9 exhibit polarized localization in the root hair tip? This information is critical for understanding the functional implications of the calcium gradient.

Response: We thank the reviewer for this concern. The subcellular localization of CNGCs in the plasma membrane at RH tips has been previously analyzed and well-established by our group and others (Tan et al., Plant Commun 2020; Brost et al, Plant J 2019). It has also been well established that the external Ca^{2+} influx is focused at the RH tips by numerous publications. We added a few sentences to introduce the widely accepted mechanism for RH growth and cited a few publications. Please see lines 133-136 and 146-147.

4. The methodology for two-electrode voltage clamp (TEVC) recordings conducted on *Xenopus* oocytes is not described in the manuscript.

Response: A description about TEVC experiments was added to the MS. Please see lines 775-789.

I look forward to seeing these revisions implemented.

Response: We thank the reviewer for the favorable comments, questions and suggestions.

REVIEWERS' COMMENTS

Reviewer #2 (Remarks to the Author):

Overall this manuscript is greatly improved.

I only have minor issues, I agree to publish this manuscript once these minor issues are addressed:

1) Line 51 -55: An early study reported that the RH interface of a two-week old winter rye (*Secale cereale*) plant is approximately 400 m², which is almost twice that of the remaining root surface. The diameter of the RH cylinder around the root is approximately ten times larger than that of the root, and thus the volume of the RH cylinder is about 100 times larger than that of the root [10].

From reference 10:

METHoDs.-Seeds were sown in culture boxes November 30, 1935, in a well lighted plant house at the

University of Iowa. The wooden containers used were 12 inches square and 22 inches deep. Each was

filled with a dark loam very lightly fertilized with a mixture of sheep manure and Vigoro. When the seedlings were well established, a centrally located plant was selected in each box and all others removed. These experiments were terminated four months later, March 30, 1936, while the plants were

growing vigorously but before they had flowered.

The surface area of all subterranean members, both roots and root hairs, of this one rye plant, was 6,875.4 square feet. This significant figure represents the potential soil contact surface of a single plant.

Wrong measurement time and wrong RH interface? 6,875 square feet = ~639 square meter. Where do the authors find the numbers for RH diameter and RH cylinder volume from this reference? These values are not implicitly stated in [10].

2)Line 755: What company supplied the [γ -32P] ATP ?

3) Lines 758 – 764

Liquid chromatography tandem mass spectrometry (LC-MS/MS)

The target sites of CPK1 in CNGC9-N were identified by LC-MS/MS as described [73]. The recombinant GST-CPK1 and CNGC9-N were incubated with 1 mM γ -18O-ATP in the reaction buffer (20 mM Tris-HCl, 10 mM MgCl₂, 1 mM DTT, and pH 7.4) at 30 °C for 1.5 h. The reaction was diluted by adding 4 volumes of 50 mM triethylammonium bicarbonate, digested with trypsin with a 1:100 (w/w) ratio, and was incubated at 37 °C overnight. After desalting, the tryptic peptides were enriched with IMAC and analyzed by LC-MS/MS.

From ref [73]:

Identification of OST1/SnRK2.6 targeting sites in CNGC-N by liquid chromatography tandem mass spectrometry (LC-MS/MS)

The recombinant GST-SnRK2.6/OST1 and the fragments of CNGC6 were incubated in reaction buffer (25 mM Tris-HCl, pH7.4, 12 mM MgCl₂, 2 mM DTT) with 1 mM γ -18O-ATP for 2 h at 25 °C. The reaction was diluted by adding 4 volumes of 50 mM triethylammonium bicarbonate, digested with trypsin with 1:100 (w/w) ratio, and incubated at 37 °C for overnight. After desalting, the tryptic peptides were enriched with IMAC and analyzed by LC-MS/MS as described (Wang et al. 2018).

All relevant manipulations and machine settings/data analysis are found in [Wang et al. 2018] and this reference should be cited, not [73]. Also, authors should add the information of relevant manipulations and machine settings/data analysis in current manuscript in method section.

Reviewer #3 (Remarks to the Author):

Thank you for addressing my concerns regarding the manuscript. However, I believe the response to my concern#1 is insufficient for the following reasons:

1. The authors did not demonstrate whether the single mutants cam1, cam2, cam3, cam4, cam5, cam6, and cam7, as well as the double mutant cam1/cam4, are true loss-of-function mutants. As I understand it, some of these mutants may not effectively disrupt the corresponding genes despite the presence of T-DNA insertions.

2. Calmodulins and calmodulin-like proteins (CMLs) likely exhibit significant redundancy. The mutants used in this study do not rule out the possibility that multiple CaMs/CMLs may redundantly regulate root hair development.

3. The authors have not provided data to support their claim that "the short root hair phenotype of the CaM7-overexpression line is a result of the artificial over-accumulation of CaM7 protein in root hairs."

To avoid delaying the publication of this work, I recommend that the authors include a brief discussion addressing concern #1. However, please do not incorporate the content added during the revision of the manuscript (Supplementary Fig. 1 and lines 184-196).

Thank you for your attention to these matters.

RESPONSES TO THE REVIEWERS

Dear reviewers,

We thank you for the valuable comments and suggestions. We revised the manuscript (MS) accordingly. In this Response-to-Reviewer file, we describe all the changes made in the MS in response to each individual reviewer concern. We provide the revised main MS file and a copy of the main MS files with all the changes highlighted in red color.

Reviewer #2 (Remarks to the Author):

Overall this manuscript is greatly improved. I only have minor issues, I agree to publish this manuscript once these minor issues are addressed:

Response: We thank for reviewer for the encouraging comments.

1) Line 51 -55: An early study reported that the RH interface of a two-week old winter rye (*Secale cereale*) plant is approximately 400 m², which is almost twice that of the remaining root surface. The diameter of the RH cylinder around the root is approximately ten times larger than that of the root, and thus the volume of the RH cylinder is about 100 times larger than that of the root [10]. From reference 10: METHoDs.-Seeds were sown in culture boxes November 30, 1935, in a well lighted plant house at the University of Iowa. The wooden containers used were 12 inches square and 22 inches deep. Each was filled with a dark loam very lightly fertilized with a mixture of sheep manure and Vigoro. When the seedlings were well established, a centrally located plant was selected in each box and all others removed. These experiments were terminated four months later, March 30, 1936, while the plants were growing vigorously but before they had flowered. The surface area of all subterranean members, both roots and root hairs, of this one rye plant, was 6,875.4 square feet. This significant figure represents the potential soil contact surface of a single plant. Wrong measurement time and wrong RH interface? 6,875 square feet = ~639 square meter. Where do the authors find the numbers for RH diameter and RH cylinder volume from this reference? These values are not implicitly stated in [10].

Response: We double checked the references, and found the number 400² is from a review paper (Rongsawt T et al., Trends Plant Sci 26(1): 83-94, 2021). According to this review paper, our description about the numbers is accurate. The review paper cited the publication by Dittmer HJ. (Am J Bot 24: 417-420, 1937) when the numbers were stated, indicating that the numbers are originally from this

research paper. We double checked the research paper, and the results are described as following in page 418 in this paper: *“The calculated total root hair surface for the entire plant was 4,321.31 square feet. This area, it will be noted, is nearly twice that of the roots. The surface area of all subterranean members, both roots and root hairs, of this one rye plant, was 6,875.4 square feet. This significant figure represents the potential soil contact surface of a single plant”*. The total root hair surface 4321.31 square feet equals to 401.43 square meter. Then the root surface should be $6875.4 - 4321.31 = 2554.1$ (square feet), which equals to 237 (square meter). Thus the total root hair surface (401 square meter) is close to twice that of root surface (237 square meter). Please note the root surface does not include the root hair surface. It seems that the numbers for the surface of root hairs and roots are correct in our description according to the review paper and the research paper, but the values in square feet from the research paper were converted into square meter in the review paper. We thus describe the root hair surface in both square feet and square meter, and cited the review paper along with the research paper (please see lines 53-54), so that readers can understand the background better.

2)Line 755: What company supplied the [γ -32P] ATP ?

Response: The [γ -32P] ATP was from a company called PerkinElmer, and the catalog number is NEG502Z500UC. This information was added to the MS (please see lines 750-751).

3) Lines 758 – 764: Liquid chromatography tandem mass spectrometry (LC-MS/MS). The target sites of CPK1 in CNGC9-N were identified by LC-MS/MS as described [73]. The recombinant GST-CPK1 and CNGC9-N were incubated with 1 mM γ -18O-ATP in the reaction buffer (20 mM Tris-HCl, 10 mM MgCl₂, 1 mM DTT, and pH 7.4) at 30 °C for 1.5 h. The reaction was diluted by adding 4 volumes of 50 mM triethylammonium bicarbonate, digested with trypsin with a 1:100 (w/w) ratio, and was incubated at 37 °C overnight. After desalting, the tryptic peptides were enriched with IMAC and analyzed by LC-MS/MS.

From ref [73]: Identification of OST1/SnRK2.6 targeting sites in CNGC-N by liquid chromatography tandem mass spectrometry (LC-MS/MS). The recombinant GST-SnRK2.6/OST1 and the fragments of CNGC6 were incubated in reaction buffer (25 mM Tris-HCl, pH7.4, 12 mM MgCl₂, 2 mM DTT) with 1 mM γ -18O-ATP for 2 h at 25 °C. The reaction was diluted by adding 4 volumes of 50 mM triethylammonium bicarbonate, digested with trypsin with 1:100 (w/w) ratio, and incubated at 37 °C for overnight. After desalting, the tryptic peptides were enriched with IMAC and analyzed by LC-MS/MS as described (Wang et al. 2018). All relevant manipulations and machine settings/data analysis are found in [Wang et al.

2018] and this reference should be cited, not [73]. Also, authors should add the information of relevant manipulations and machine settings/data analysis in current manuscript in method section.

Response: We thank the reviewer for pointing this error out. The reference Wang et al., 2018 was cited, and reference [73] was removed as suggested (please see line 755). More detailed relevant information about manipulations, machine settings and data analysis was added to the MS (please see lines 760-771).

Reviewer #3 (Remarks to the Author):

Thank you for addressing my concerns regarding the manuscript. However, I believe the response to my concern#1 is insufficient for the following reasons:

Response: We thank the reviewer for the professional review of our MS.

1. The authors did not demonstrate whether the single mutants *cam1*, *cam2*, *cam3*, *cam4*, *cam5*, *cam6*, and *cam7*, as well as the double mutant *cam1/cam4*, are true loss-of-function mutants. As I understand it, some of these mutants may not effectively disrupt the corresponding genes despite the presence of T-DNA insertions.

Response: We thank the reviewer for the comments. The *cam* mutants have been identified as knockout mutants by RT-qPCR experiments previously (Niu WT et al., J Exp Bot 71, 90-104, 2020). Based on the comments and suggestions of both reviewers and editors, we removed the *cam* data and the related description from this article.

2. Calmodulins and calmodulin-like proteins (CMLs) likely exhibit significant redundancy. The mutants used in this study do not rule out the possibility that multiple CaMs/CMLs may redundantly regulate root hair development.

Response: We agree with the reviewer that we can't completely exclude the involvement of calmodulins in RH growth. The lack of RH phenotype in *cam* mutants may be a result of functional redundancy between calmodulins. According to the comments and suggestions from both reviewers and editors, we removed the data about the *cam* mutants and the description from this article. Thank you for your concern.

3. The authors have not provided data to support their claim that "the short root hair phenotype of the CaM7-overexpression line is a result of the artificial over-accumulation of CaM7 protein in root hairs." To avoid delaying the publication of this work, I recommend that the authors include a brief discussion addressing concern #1. However, please do not incorporate the content added during the revision of the manuscript (Supplementary Fig. 1 and lines 184-196).

Response: This statement "the short root hair phenotype of the CaM7-overexpression line is a result of the artificial over-accumulation of CaM7 protein in root hairs" is a personal speculation. According to the comments and suggestions of both reviewers and editors, we removed the *cam* mutant data (Supplementary Figure 1) and the description in the main MS from this article. Thank you for your comments.

Thank you for your attention to these matters.

Response: Thank you for your favorable comments and valuable suggestions in the two rounds of review.